# Exploring the Entropy Mechanism in On-policy Optimization for LLM Agents

## Abstract

Training LLM agents in multi-turn environments with sparse rewards, where completing a single task requires 30+ turns of interaction within an episode, presents a fundamental challenge for reinforcement learning. We identify a critical failure mode unique to this setting: the exploration-exploitation cascade failure. This cascade begins with early-stage policy premature convergence, where sparse feedback causes agents to commit to flawed, low-entropy strategies. Subsequently, agents enter late-stage policy collapse, where conventional entropy regularization becomes counterproductive, promoting chaotic exploration that destabilizes training. We propose Entropy-regularized Policy Optimization (EPO), a general framework that breaks this failure cycle through three synergistic mechanisms: (1) adopting entropy regularization in multi-turn settings to enhance exploration, (2) an entropy smoothing regularizer that bounds policy entropy within historical averages to prevent abrupt fluctuations, and (3) adaptive phase-based weighting that balances exploration and exploitation across training. Our analysis validates that EPO mitigates entropy variance through smoothing regularizers, which suppresses the oscillations. EPO achieves up to 152% performance improvement on ScienceWorld and up to 19.8% on ALFWorld. Our work demonstrates that multi-turn sparse-reward settings require fundamentally different entropy control than traditional RL, with broad implications for LLM agent training. The code is available at the URL [1].

## 1 Introduction

Reinforcement Learning (RL) (Shao et al., 2024; Schulman et al., 2017) has become an important approach for post-training Large Language Models (LLMs), enabling LLMs to acquire reasoning ability with single-turn verified reward (Guo et al., 2025; Yang et al., 2025). LLM agents start to achieve success in versatile scenarios, such as coding (Anthropic, 2025; Kipruto & Salva, 2025; OpenAI, 2025b; Wang et al., 2025b), computing using (OpenAI, 2025a; Xie et al., 2024; Zheng et al., 2024), and searching (Team et al., 2025; Zeng et al., 2025), benefiting from LLM's powerful reasoning ability. Multi-turn LLM agent scenarios present unique challenges compared to traditional LLM RL settings, as episodes can span 30+ turns (Shridhar et al., 2021; Wang et al., 2022) with extended trajectories and sparse rewards. Unlike conventional RL, where rewards are distributed throughout the episode, LLM agents typically receive feedback only at completion, meaning the policy is updated based on the shared reward across long-horizon turns within an episode, making the exploration-exploitation trade-off (Sutton, 1988) crucial for stable training.

Traditional RL approaches (Haarnoja et al., 2018; Mnih et al., 2016; Williams & Peng, 1991) employ entropy regularization by adding the entropy of the policy to the objective function, discouraging premature convergence to suboptimal deterministic policies through a hyperparameter-controlled regularization term. In RL training for LLM, researchers have adapted these mechanisms by incorporating entropy into the advantage function to reward high-entropy token generation (Cui et al., 2025; Dong et al., 2025; He et al., 2025; Wang et al., 2025a), encouraging exploration and preventing entropy collapse in later training stages. However, through extensive empirical analysis on standard multi-turn agent benchmarks (ScienceWorld and ALFWorld), we uncover a critical, previously unidentified limitation in existing methods. We observe that applying standard entropy-controlled baselines to

---

[1] https://anonymous.4open.science/r/EPO-E637/

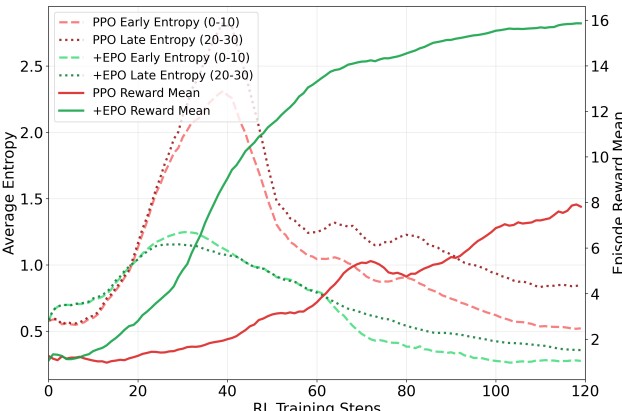

Figure 1: The exploration-exploitation cascade failure observed in standard PPO. We identify two distinct failure phases. During Phase 1 of Excessive Early Exploration (steps 0 to 40) the PPO early trajectory steps (pink dashed line) exhibit uncontrolled entropy growth and stagnant rewards. This triggers Phase 2 of Uncertainty Propagation (steps 40 to 120) where instability cascades to late trajectory steps (red dotted line) causing persistent high entropy and reward plateaus. In contrast, our EPO method maintains stable, controlled entropy levels across both early and late trajectory steps throughout training, achieving significantly lower final entropy values and consistent reward improvement, preventing the cascade failure.

these established tasks triggers a catastrophic failure pattern we term *the exploration-exploitation cascade failure*. As illustrated in Figure 1, this phenomenon consistently manifests in two distinct phases on standard benchmarks: In Phase 1 (0-40 steps), standard entropy regularization paradoxically causes *excessive early-stage exploration* with uncontrolled entropy growth (from 0.5 to 2.5), creating unstable behavioral foundations while rewards remain stagnant. In Phase 2 (40-120 steps), this early instability cascades into *late-stage uncertainty propagation*, where dangerously high entropy oscillations persist (1.0-2.0), preventing coherent strategy formation and causing reward plateaus. In contrast, our EPO method maintains stable, controlled entropy levels (declining from 1.2 to 0.3) throughout training, achieving consistent reward improvement from 2 to 16. This cascade failure is not an artifact of specific synthetic settings, but a fundamental failure mode that emerges naturally when standard RL objectives interact with multi-turn sparse reward environments. Therefore, *how to design exploration mechanisms that prevent this cascade failure while maintaining necessary exploration* remains an important open problem for multi-turn agent training.

To address this cascade issue, we propose Entropy-regularized Policy Optimization (**EPO**), a novel framework that combines entropy regularization with specialized mechanisms for stable on-policy training under sparse reward conditions. Our framework is guided by the central insight that standard entropy regularization is insufficient because it lacks temporal awareness. We find that anchoring the policy's entropy to a dynamically adjusted historical bound provides the necessary stability to halt the exploration-exploitation cascade failure without sacrificing essential exploration.

Our contributions are fourfold. ❶ *We empirically discover and formally characterize the exploration-exploitation cascade failure on standard multi-turn benchmarks*, demonstrating that this novel failure mode—where excessive early-stage exploration compounds into late-stage uncertainty—is the primary cause of poor performance in standard RL baselines. ❷ *We adapt entropy regularization to the multi-turn setting by computing entropy across all turns within trajectories and averaging over trajectory batches*, providing a foundation that captures the unique temporal structure of agent interactions. ❸ *We introduce an entropy smoothing regularizer that penalizes deviations from historical entropy averages*, effectively dampening the severe oscillations between overconfidence and over-exploration that characterize the cascade failure. ❹ *We develop an adaptive weighting scheme that dynamically balances exploration and exploitation across training phases*, starting with conservative exploration, transitioning through balanced exploration-exploitation, and ending with strong stabilization to ensure convergence. Together, these components create a theoretically grounded and general framework that prevents the empirically observed cascade failure and ensures optimal exploration-exploitation trade-offs while being compatible with any on-policy optimization method. We validate EPO on challenging benchmarks ScienceWorld (Wang et al., 2022) and ALFWorld (Shridhar et al., 2021), achieving up to 152% performance improvement with significantly

more stable training dynamics, transforming previously untrainable sparse-reward scenarios into smoothly converging optimization problems.

## 2 RELATED WORK

### 2.1 REINFORCEMENT LEARNING FOR LLMs

RLHF (Ouyang et al., 2022) and DPO (Rafailov et al., 2023) have become foundational approaches for aligning LLMs with human preferences, with both methods significantly improving model alignment and instruction-following capabilities. Recent RL methods such as GRPO (Shao et al., 2024) and DAPO (Yu et al., 2025) further enhance LLM reasoning abilities during post-training through verified rewards. In contrast to PPO (Schulman et al., 2017), these methods leverage batch-wise advantage computation from identical prompts, obviating the critic model and substantially improving the computational tractability of large-scale RL training for LLMs. However, a primary challenge in applying RL to LLMs is the phenomenon of policy entropy collapse: models rapidly reduce their stochasticity, converging on narrow, over-optimized behaviors (Cui et al., 2025; Dong et al., 2025; Wang et al., 2025a; Deng et al., 2025; He et al., 2025; Cheng et al., 2025a;b). In response, recent work has focused on integrating entropy control mechanisms into the optimization process to preserve policy diversity. For instance, Cui et al. (Cui et al., 2025) regulate the impact of high-covariance tokens by applying a clipping function and a KL penalty. Meanwhile, Cheng et al. (Cheng et al., 2025b) augment the standard advantage function with a clipped, gradient-detached entropy term, which encourages deeper reasoning chains without altering the original policy optimization direction.

### 2.2 REINFORCEMENT LEARNING FOR LLM AGENTS

To enhance their autonomy, LLM agents are designed to interact with external environments using diverse toolsets (OpenAI, 2025a; Team, 2025). However, training agents to complete multi-step tasks with these tools presents significant challenges for standard reinforcement learning, including sparse rewards and credit assignment problems. To address these issues, seminal works have introduced advanced training paradigms such as hierarchical RL (Zhou et al., 2024), autonomous learning (Bai et al., 2024), and off-policy Q-learning (Bai et al., 2025). Meanwhile, another line of research employs supervised fine-tuning (SFT) to directly enhance the models' decision-making abilities, training them on vast datasets of high-quality tool-use trajectories to master complex environments and APIs (Xi et al., 2024; Zhang et al., 2024; Qin et al., 2024). Recently, to leverage the training stability of GRPO (Shao et al., 2024), researchers have extended single-turn GRPO to multi-turn training settings with various training techniques to improve performance (Jin et al., 2025; Feng et al., 2025; Wang et al., 2025c). To guide learning in long-horizon scenarios, RLVMR (Zhang et al., 2025) introduces verifiable meta-reasoning rewards that provide dense, intermediate feedback on the agent's reasoning process. However, these methods overlook the critical exploration-exploitation dynamics unique to multi-turn settings, failing to address the exploration-exploitation cascade failure where policies undergo early-stage premature convergence followed by late-stage excessive exploration with high entropy that makes training vulnerable to noise. This fundamental gap leads to unstable training dynamics and prevents agents from effectively learning long-horizon strategies.

## 3 PRELIMINARY

### 3.1 ON-POLICY OPTIMIZATION

On-policy optimization is a fundamental paradigm in reinforcement learning where the agent learns to improve its policy by directly optimizing the expected return using trajectories sampled from the current policy. Given a parameterized stochastic policy $\pi_\theta(a|s)$ with parameters $\theta \in \mathbb{R}^d$, the objective is to maximize the expected return, given by $J(\theta) = \mathbb{E}_{\tau \sim \pi_\theta}[R(\tau)]$, where $\tau$ denotes a trajectory and $R(\tau) = \sum_{t=0}^{T} \gamma^t r_t$ is the discounted return. On-policy optimization methods build on:

$$\nabla_\theta J(\theta) = \mathbb{E}_{\tau \sim \pi_\theta} \left[ \sum_{t=0}^{T} \nabla_\theta \log \pi_\theta(a_t|s_t) A^{\pi_\theta}(s_t, a_t) \right] \quad (1)$$

where $A^{\pi_\theta}(s_t, a_t) = Q^{\pi_\theta}(s_t, a_t) - V^{\pi_\theta}(s_t)$ is the advantage function. While the policy gradient provides an unbiased estimator of $\nabla_\theta J(\theta)$, directly optimizing this objective can lead to instability due to large policy updates. To address this, modern on-policy methods employ surrogate objective functions that approximate the policy gradient while ensuring stable learning. The standard policy gradient can be reformulated as the surrogate objective $L^{PG}(\theta) = \mathbb{E}_{\tau \sim \pi_{\theta_{\text{old}}}} \left[ \frac{\pi_\theta(a_t|s_t)}{\pi_{\theta_{\text{old}}}(a_t|s_t)} \hat{A}_t \right]$. However, this surrogate objective is only valid for infinitesimally small updates. Proximal Policy Optimization (PPO) (Schulman et al., 2017) addresses this limitation by constraining the policy ratio through a clipped surrogate objective:

$$L^{CLIP}(\theta) = \mathbb{E}_{\tau \sim \pi_{\theta_{\text{old}}}} \left[ \min \left( r_t(\theta) \hat{A}_t, \text{clip}(r_t(\theta), 1 - \epsilon, 1 + \epsilon) \hat{A}_t \right) \right] \tag{2}$$

where $r_t(\theta) = \frac{\pi_\theta(a_t|s_t)}{\pi_{\theta_{\text{old}}}(a_t|s_t)}$ is the importance sampling ratio, $\mathbb{E}_{\tau \sim \pi_{\theta_{\text{old}}}}[\cdot]$ denotes expectation over trajectories sampled under $\pi_{\theta_{\text{old}}}$, and $\epsilon$ defines the trust region bounds. Group Relative Policy Optimization (GRPO) (Shao et al., 2024) extends PPO by modifying the advantage function computation. Instead of using standard advantages $\hat{A}_t$, GRPO employs group-relative advantages, computed as $\tilde{A}_t = \frac{R_t - \mu_g}{\sigma_g + \delta}$. where $R_t$ is the return from timestep $t$, $\mu_g$ and $\sigma_g$ are the mean and standard deviation of returns within group $g$, and $\delta$ is a small constant for numerical stability. This group-based normalization provides more stable gradient estimates.

## 3.2 Problem Formulation

We formalize the multi-turn task as a sequential decision-making reinforcement learning problem. A single LLM agent $\pi_\theta$ executes a task through $T$ turns over a trajectory $\tau = (s_0, a_0, r_0, \ldots, s_T, a_T, r_T)$. The reward is sparse, with $r_t = 0$ for all intermediate turns and only the final turn receiving the task outcome reward, such that the total return is $R(\tau) = \sum_{t=0}^{T} \gamma^t r_t = r_T$, where $r_T \in \{0, 1\}$ represents the binary task outcome. In our experimental settings, specifically ALFWorld (Shridhar et al., 2021) and SciWorld (Wang et al., 2022), we assume an undiscounted formulation where $\gamma = 1$. All turns within the same task share the final outcome reward, creating a credit assignment challenge across sequential turns. The multi-turn policy optimization differs from standard RL in that losses accumulate across all $T$ turns before parameter updates:

$$L^{MT}(\theta) = \mathbb{E}_{\tau \sim \pi_{\theta_{\text{old}}}} \left[ \mathbb{E}_{t \sim T} \left[ \min \left( r_t(\theta) A_t, \text{clip}(r_t(\theta), 1 - \epsilon, 1 + \epsilon) A_t \right) \right] \right] \tag{3}$$

where $r_t(\theta) = \frac{\pi_\theta(a_t|s_t)}{\pi_{\theta_{\text{old}}}(a_t|s_t)}$ and $A_t$ represents the advantage estimate at turn $t$ within the multi-turn trajectory. $\mathbb{E}_t[\cdot]$ denotes expectation over timesteps within turn $t$, while the outer expectation is over trajectories sampled under $\pi_{\theta_{\text{old}}}$.

# 4 Methodology

## 4.1 Entropy Regularization

To address the exploration-exploitation cascade failure, we first adapt entropy regularization to capture the temporal structure of multi-turn interactions. Unlike traditional RL where entropy is computed per-step, we recognize that in multi-turn environments, early trajectory decisions compound across subsequent turns. Therefore, we compute entropy across all turns within each trajectory and average over the batch of trajectories. The entropy-regularized policy loss is formulated as $L^{ER}(\theta) = L^{MT}(\theta) - \lambda L^H(\theta)$, where $\lambda$ is the entropy coefficient, and the entropy loss is averaged over the batch of trajectories:

$$L^H(\theta) = \frac{1}{B} \sum_{j=1}^{B} \frac{1}{T} \sum_{t=0}^{T-1} \frac{1}{|\tau_{j,t}|} \sum_{i=1}^{|\tau_{j,t}|} \mathcal{H}_{j,t,i} \tag{4}$$

where $B$ is the batch size (number of trajectories), $T$ is the number of turns per trajectory, $\mathcal{H}_{j,t,i}$ is the entropy at token position $i$ in turn $t$ of trajectory $j$, and $|\tau_{j,t}|$ represents the sequence length at turn $t$ of trajectory $j$. The token-level entropy $\mathcal{H}$ is computed from the model's probability distribution over the vocabulary at each position as $\mathcal{H} = -\sum_{v \in V} p(v|w_{<t}) \log p(v|w_{<t})$, where $p(v|w_{<t})$ is the probability of token $v$ from vocabulary $V$ given the preceding context $w_{<t}$.

## 4.2 ENTROPY SMOOTHING REGULARIZER

To break the cascade failure's two-phase pattern, excessive early exploration followed by late-stage uncertainty propagation, we introduce an entropy smoothing mechanism that prevents the dangerous oscillations observed in sparse reward settings. We maintain an entropy history window $\mathcal{W}_k = \{\bar{H}_0, \ldots, \bar{H}_m, \ldots, \bar{H}_{k-1}\}$ for RL step $k$, storing the average entropy $\bar{H}_m$ across all trajectories at the token level for each previous RL step $m$. The historical entropy reference is computed as the mean $\bar{H}^{\mathcal{W}_k} = \frac{1}{k} \sum_{m=0}^{k-1} \bar{H}_m$. This historical anchoring prevents the uncontrolled entropy growth that characterizes the early exploration phase of cascade failure. We apply a token-wise penalty based on acceptable entropy ranges relative to this historical average:

$$\mathcal{P}_{n,t,i} = \alpha \cdot \left[ \max\left(0, \kappa_l \bar{H}^{W_k} - H_{n,t,i}\right) + \max\left(0, H_{n,t,i} - \kappa_r \bar{H}^{W_k}\right) \right] \tag{5}$$

The boundary coefficients $\kappa_l$ and $\kappa_r$ define the acceptable range, while $\alpha$ provides the penalty weight for tokens with entropy outside the desired range. By bounding entropy within historical averages, we prevent both the blind exploration of early stages and the chaotic uncertainty propagation of late stages. Aggregating these penalties across all tokens, turns, and trajectories yields the smoothing loss:

$$L^{smooth}(\theta) = \frac{1}{B} \sum_{n=1}^{B} \frac{1}{T} \sum_{t=0}^{T-1} \frac{1}{|\tau_{n,t}|} \sum_{i=1}^{|\tau_{n,t}|} \mathcal{P}_{n,t,i} \tag{6}$$

The complete entropy-smoothed policy optimization loss is then defined as $L^{EPO}(\theta) = L^{MT}(\theta) - \lambda[L^H(\theta) - \beta_k L^{smooth}(\theta)]$, where the dynamic coefficient $\beta_k$ follows an exponential schedule that directly counteracts the cascade failure's progression:

$$\beta_k = \begin{cases} \beta_{start} + (1 - \beta_{start})\left(1 - e^{-\lambda_d \frac{k}{k_{mid}}}\right), & \text{if } k \le k_{mid} \\ 1 + (\beta_{end} - 1)\left(1 - e^{-\lambda_d \frac{k - k_{mid}}{K - k_{mid}}}\right), & \text{if } k > k_{mid} \end{cases} \tag{7}$$

This adaptive schedule specifically addresses each phase of the cascade failure: it begins with conservative exploration to prevent early-stage behavioral lock-in, transitions through balanced exploration-exploitation at mid-training ($k_{mid} = \lfloor K/2 \rfloor$), and increases smoothing strength in later phases to prevent uncertainty propagation and ensure stable convergence. The parameters $\beta_{start}$, $\beta_{end}$, $\lambda_d$, and $K$ control the transition dynamics and training duration.

Algorithm 1 presents the full optimization procedure, incorporating these components.

## 5 EXPERIMENTS

### 5.1 EXPERIMENTS SETUP

This section outlines our experimental setup, including the benchmarks, evaluation protocol, baselines, and implementation details. Further details can be found in Appendix B.

**Benchmark.** We evaluate on two challenging benchmarks that require different reasoning capabilities, ScienceWorld (Wang et al., 2022) and ALFWorld (Shridhar et al., 2021). ScienceWorld focuses on text-based scientific experimentation, demanding systematic hypothesis testing and structured exploration. ALFWorld is an embodied environment containing 4,639 household task instances across six categories, requiring multi-step decision-making and spatial reasoning. To improve the generalizability of our approach across multiple scenarios, we finetune the foundation model directly on the environment using RL, rather than employing trajectory finetuning for initialization.

**Evaluation Setting.** To evaluate generalization capabilities, we focus on two key evaluation scenarios: **IID** (in-distribution) covers seen task variants and categories, while **OOD** (out-of-distribution) evaluates on unseen task variants within seen categories. This design allows us to measure both optimization effectiveness and generalization robustness, which are crucial for practical deployment. We employ dual success rate metrics to capture different aspects of performance: **Succ.**$^*$ reports the average of maximum success rates across random seeds, while $\overline{\textbf{Succ.}}$ measures average performance

---

**Algorithm 1** Entropy-smoothed Policy Optimization (EPO)

---

**Require:** Initial policy parameters $\theta$, entropy coefficient $\lambda$, smoothing penalty $\alpha$, entropy bounds $\kappa_l, \kappa_r$, and schedule parameters for $\beta_k$.
 1: Initialize policy parameters $\theta_0$.
 2: Initialize historical entropy window $\mathcal{W}_0 = \emptyset$.
 3: **for** each training step $k = 0, 1, 2, \ldots, K - 1$ **do**
 4:     Collect a batch of $B$ trajectories $\mathcal{D}_k = \{\tau_j\}_{j=1}^B$ using the old policy $\pi_{\theta_{\text{old}}}$.
 5:     Compute advantages $\hat{A}_t$ for all timesteps in $\mathcal{D}_k$.
 6:     Compute the multi-turn policy loss $L^{MT}(\theta)$ using Equation 3.
 7:                                        ▷ — *Start of EPO specific computations* —
 8:     Compute the trajectory-aware entropy loss $L^H(\theta)$ over the batch using Equation 4.
 9:     **if** $k > 0$ **then**
10:         Compute the historical entropy mean $\bar{H}^{\mathcal{W}_k}$ from the window $\mathcal{W}_k$.
11:         Compute the smoothing loss $L^{\text{smooth}}(\theta)$ based on $\bar{H}^{\mathcal{W}_k}$ using Equation 6.
12:     **else**
13:         $L^{\text{smooth}}(\theta) \leftarrow 0$.
14:     **end if**
15:     Update the dynamic coefficient $\beta_k$ according to the schedule in Equation 7.
16:     Combine losses to form the final EPO objective:
17:     $L^{EPO}(\theta) = L^{MT}(\theta) - \lambda[L^H(\theta) - \beta_k L^{\text{smooth}}(\theta)]$.
18:     Update the policy parameters $\theta$ by optimizing $L^{EPO}(\theta)$.
19:     Update the historical entropy window $\mathcal{W}_{k+1} \leftarrow \mathcal{W}_k \cup \{\bar{H}_{\text{current batch}}\}$.
20: **end for**
**Ensure:** Optimized policy parameters $\theta_K$.

---

after convergence, reflecting practical reliability. Given the high variance inherent in RL, final performance scores alone can be misleading. We therefore present averaged curves to provide a more robust comparison and illustrate the performance evolution throughout the training process.

**Baselines.** We conduct comprehensive comparisons across multiple paradigms: (1) **Prompting-based approaches** such as ReAct (Yao et al., 2023) that utilize in-context learning without parameter optimization; (2) **Trajectory-based and platform methods** including supervised fine-tuning (SFT) through expert trajectory imitation and AgentGym (Xi et al., 2024) which provides a unified framework with behavioral cloning and self-evolution mechanisms; (3) **General reinforcement learning approaches** encompassing standard on-policy methods (PPO (Schulman et al., 2017), GRPO (Shao et al., 2024)); (4) **Agent RL approaches** including recent methods specifically designed for agent training (GiGPO (Feng et al., 2025), RLVMR (Zhang et al., 2025)). Our proposed EPO methodology is architected as a general enhancement framework that can be seamlessly integrated with existing RL paradigms, as exemplified through our PPO+EPO and GRPO+EPO implementations.

**Implementation Details.** To optimize performance for each benchmark's complexity, we employ Qwen2.5-3B-Instruct for ALFWorld and the larger Qwen2.5-7B-Instruct for ScienceWorld's more complex scientific reasoning tasks. Constrained by computational resources, we adopt a single foundation model per task whose size is sufficient to ensure proper convergence, as smaller models consistently fail to converge. We conduct our own implementations and experiments for our proposed method, PPO, and GRPO baselines across three random seeds to ensure statistical reliability. Results for other baseline methods (ReAct, AgentGym, SFT, GiGPO, RLVMR) are sourced from the RLVMR's (Zhang et al., 2025) paper. To account for their different convergence characteristics, we trained the model for 120 RL steps on ScienceWorld and 150 steps on ALFWorld.

## 5.2 PERFORMANCE COMPARISON

**Quantitative Results Analysis.** Table 1 presents comprehensive performance comparisons across both ScienceWorld and ALFWorld environments. Our EPO enhancement demonstrates substantial improvements when integrated with existing RL methods. Notably, PPO+EPO achieves a remarkable 152.1% improvement in averaged success rates ($\overline{Succ.}$) on ScienceWorld IID tasks, significantly outperforming agent-specialized RL methods including GiGPO(53.4%) and RLVMR(67.2%). The dramatic improvement of PPO stems from addressing its inherent tendency toward aggressive policy

Table 1: For PPO and GRPO baselines with our EPO method: **better performance** indicates improvement over baseline, **worse performance** indicates degradation. Highlighted values represent the best performance among other baseline methods. $\Delta$ shows the relative improvement (%) when applying our method. Results for other baseline methods (ReAct, AgentGym, SFT, GiGPO, RLVMR) are sourced from the RLVMR (Zhang et al., 2025) paper. We ran our own implementations of PPO, GRPO, and EPO, tuning hyperparameters across multiple trials to obtain stable results.

| Method | LLM | ScienceWorld | | | | LLM | ALFWorld | | | |
| | | IID | | OOD | | | IID | | OOD | |
| | | Succ.* | $\overline{Succ.}$ | Succ.* | $\overline{Succ.}$ | | Succ.* | $\overline{Succ.}$ | Succ.* | $\overline{Succ.}$ |
| ReAct | GPT-4o | 45.4 | - | 49.2 | - | GPT-4o | 57.3 | - | 66.0 | - |
| ReAct | DeepSeek-R1 | 22.2 | - | 31.4 | - | DeepSeek-R1 | 68.8 | - | 70.2 | - |
| ReAct | Qwen2.5-7B | 7.8 | - | 11.3 | - | Qwen2.5-7B | 23.1 | - | 28.5 | - |
| AgentGym | LLaMa2-7B | 46.9 | - | 33.6 | - | LLaMa2-7B | 76.6 | - | 63.3 | - |
| SFT | Qwen2.5-7B | 36.7 | - | 32.0 | - | Qwen2.5-7B | 63.3 | - | 57.0 | - |
| GiGPO | Qwen2.5-7B | 53.4 | - | 35.2 | - | Qwen2.5-7B | 89.5 | - | 90.2 | - |
| RLVMR | Qwen2.5-7B | 67.2 | - | 43.0 | - | Qwen2.5-7B | 91.4 | - | 91.8 | - |
| PPO | Qwen2.5-7B | 64.6 | 38.4 | 58.3 | 39.1 | Qwen2.5-3B | 95.8 | 72.3 | 87.5 | 70.9 |
| +EPO | Qwen2.5-7B | **100.0** | **96.8** | **100.0** | **96.2** | Qwen2.5-3B | 85.4 | **73.4** | **91.7** | **74.3** |
| $\Delta$ | | **54.8%** | **152.1%** | **71.5%** | **146.0%** | | **-10.9%** | **1.5%** | **4.8%** | **4.8%** |
| GRPO | Qwen2.5-7B | 93.8 | 81.6 | 91.7 | 80.9 | Qwen2.5-3B | 87.5 | 63.3 | 83.3 | 63.5 |
| +EPO | Qwen2.5-7B | **95.8** | **83.8** | **95.8** | **81.3** | Qwen2.5-3B | **91.7** | **75.8** | **89.6** | **75.4** |
| $\Delta$ | | **2.1%** | **2.7%** | **4.5%** | **0.5%** | | **4.8%** | **19.8%** | **7.6%** | **18.7%** |

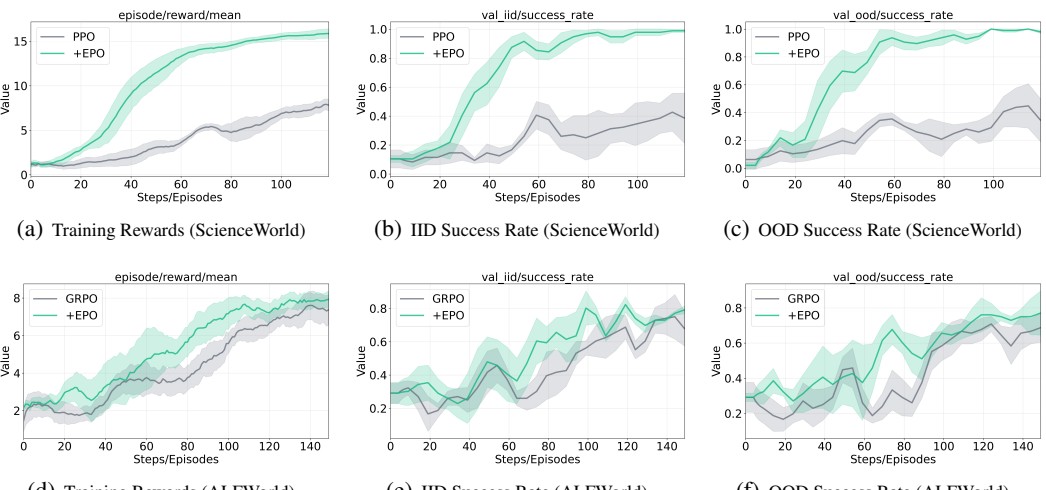

(a) Training Rewards (ScienceWorld)   (b) IID Success Rate (ScienceWorld)   (c) OOD Success Rate (ScienceWorld)

(d) Training Rewards (ALFWorld)   (e) IID Success Rate (ALFWorld)   (f) OOD Success Rate (ALFWorld)

Figure 2: Training dynamics and generalization performance analysis. We present the evolution of training rewards and validation success rates across both in-distribution (IID) and out-of-distribution (OOD) evaluation settings. (a-c) ScienceWorld experimental results contrasting PPO and PPO+EPO performance across training reward accumulation, IID validation, and OOD validation metrics. (d-f) ALFWorld experimental results contrasting GRPO and GRPO+EPO under identical evaluation criteria. Our EPO enhancement exhibits significantly improved training stability and substantial performance gains across both IID and OOD evaluation scenarios against baseline methods.

updates that can cause severe entropy collapse in multi-turn interactions, particularly problematic in ScienceWorld's sparse reward environment where maintaining exploration is crucial. Our entropy smoothing regularization directly mitigates this issue by stabilizing policy entropy across sequential turns. In contrast, GRPO's more conservative update mechanism makes it less susceptible to such entropy instabilities, resulting in more modest but consistent improvements (9.5% on ALFWorld IID tasks) when combined with EPO. We emphasize the $\overline{Succ.}$ metric as it represents performance averaged across multiple evaluation episodes, providing a more robust and fair comparison by reducing variance from individual runs.

**Training Dynamics Analysis.** Figure 2 illustrates the training dynamics and validation performance, revealing key insights into our method's effectiveness. The training reward curves demonstrate that EPO-enhanced methods achieve substantially higher reward accumulation while maintaining

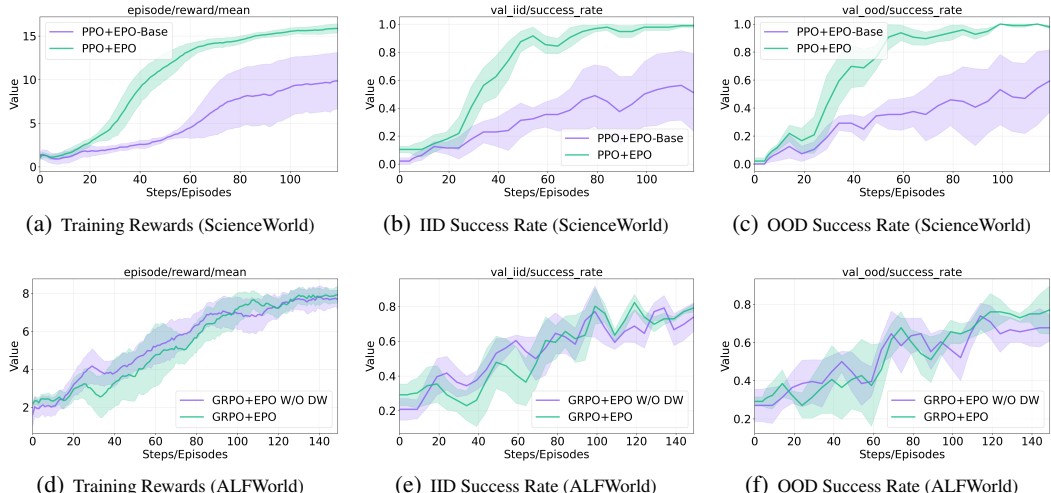

Figure 3: Ablation studies on entropy regularization components. **(a-c)** ScienceWorld comparison of EPO versus EPO-Base without entropy smoothing, demonstrating that smoothing is essential for stable convergence in sparse reward settings. **(d-f)** ALFWorld comparison of EPO with dynamic $\beta_k$ versus EPO W/O DW using constant $\beta$, showing that adaptive weighting significantly accelerates early training progress.

superior stability. In ScienceWorld, PPO+EPO reaches approximately 2x higher training rewards (15 vs. 8) with smooth monotonic trajectories, while ALFWorld shows consistent improvements with GRPO+EPO maintaining steady upward trends throughout training. The validation performance curves reveal the most significant improvements in early training phases and overall convergence behavior. On ScienceWorld, our EPO variants achieve rapid convergence to high success rates (>0.8 for both IID and OOD) within 40 training steps, compared to baseline methods that struggle to exceed 0.4 even after 100 steps. On ALFWorld, while baseline methods exhibit substantial oscillations and instability, our EPO-enhanced approaches demonstrate more consistent performance with reduced variance, particularly evident in the OOD evaluation where baselines frequently drop below 0.2 while EPO variants maintain performance above 0.4. The key observable pattern across both environments is the elimination of the characteristic oscillations between premature convergence and over-exploration that plague standard RL methods. This stability translates to more reliable training outcomes and enhanced generalization capabilities, directly validating our entropy regularization framework's effectiveness in addressing the exploration-exploitation dilemma in multi-turn LLM agent training.

## 5.3 ABLATION STUDY

Figure 3 presents ablations on two key components: the entropy smoothing regularizer and its dynamic weighting coefficient $\beta_k$. On ScienceWorld, removing the smoothing regularizer (EPO-Base) severely degrades performance. EPO-Base exhibits delayed convergence with minimal rewards until step 40 and plateaus at 0.5-0.6 success rate, while full EPO achieves meaningful learning by step 20 and reaches 0.8-1.0 success rate, demonstrating a 50-60% improvement. ScienceWorld's sparse reward structure induces severe oscillations between exploration and exploitation, which the smoothing regularizer effectively stabilizes by maintaining entropy within historical bounds defined by $\kappa_l$ and $\kappa_r$. On ALFWorld, replacing the dynamic coefficient $\beta_k$ with a constant weight (EPO W/O DW) yields similar final performance around 0.7-0.8 success rate but results in slower convergence during the initial 40-60 episodes and increased training variance. The adaptive schedule in Equation 7, which progresses from $\beta_{start}$ through unity to $\beta_{end}$, automatically modulates regularization intensity across training phases, thereby improving convergence efficiency while preserving final performance quality. For more detailed ablation study, it can refer to Appendix B.3.

## 5.4 MODEL STUDY

We compare consistent entropy regularization (PPO+EPO-Base) against a decaying schedule (PPO+EPO-Decay). This approach dynamically adjusts the exploration-exploitation balance by

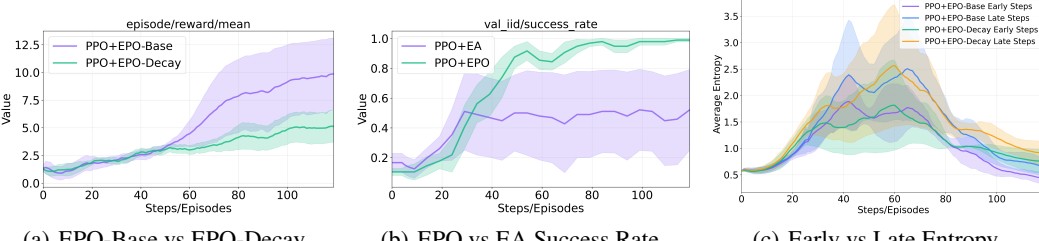

(a) EPO-Base vs EPO-Decay     (b) EPO vs EA Success Rate     (c) Early vs Late Entropy

Figure 4: Model studies on ScienceWorld employing PPO with weighted entropy loss and entropy-based advantage shaping (Cheng et al., 2025b).

applying a high weight to the entropy loss in early steps and a progressively smaller weight in later steps to promote exploitation. Counter-intuitively, Figure 4(a) shows that decay consistently underperforms by prematurely suppressing early-turn exploration, locking agents into suboptimal strategies. We also compare EPO against the Entropy-based Advantage (EA) method from (Cheng et al., 2025b), Figure 4(b) shows EPO converges to near-perfect success rates ($\sim$1.0) while EA plateaus at 0.5–0.6. Unlike EA's detached entropy terms, EPO provides direct gradient signals $\nabla_\theta L^H(\theta)$ and temporal consistency through historical entropy smoothing. These studies reveal two key insights: (1) multi-turn sparse-reward tasks require sustained exploration rather than conventional exploration-to-exploitation scheduling, and (2) direct entropy optimization with temporal smoothing is superior to indirect advantage shaping for maintaining long-horizon policy stability. See Appendix B.4 for detailed analysis.

## 6 THEORETICAL ANALYSIS

Our theoretical analysis shows that EPO achieves superior performance bounds by introducing a corrective mechanism that reduces the substantial bias of standard maximum entropy RL. This correction is driven by a "smoothing gap," $\Phi$, which measures the entropy instability of the current policy relative to an optimal one. This addresses a key limitation of prior methods, which control *whether* to explore but not *how to explore stably*. By using an adaptive coefficient $\beta_k$ to stabilize exploration, EPO resolves the entropy oscillation phenomenon observed in practice. The complete proof is provided in Appendix A.

**Proposition 1** (Improved Performance Bound with EPO). *Assume the policy follows a softmax parameterization. If the EPO policy gradient satisfies $\|\nabla V_{\lambda,\beta}^{\pi_\theta}(\mathcal{D})\| \leq \epsilon$, then for any query $s_0$, the policy suboptimality is bounded by:*

$$V^{\pi^*}(s_0) - V^{\pi_\theta}(s_0) \leq \underbrace{\frac{|\mathcal{D}|^2}{C_{\lambda,\beta}^{\pi_\theta}(s_0)}\frac{\epsilon^2}{2\lambda}}_{\text{Optimization Error}} + \underbrace{\lambda H \log \frac{|\mathcal{A}|}{|\mathcal{A}_H^*(s_0)|^{\frac{1}{H}}}}_{\text{Standard Entropy Bias}} - \underbrace{\lambda\beta_k \Phi^{\pi_\theta,\pi^*}(s_0)}_{\text{Corrective Bias from Smoothing}}$$

*where:*

- $\Phi^{\pi_\theta,\pi^*}(s_0) := \mathbb{E}_{\pi_\theta}[L^{smooth}(\theta)|s_0] - \mathbb{E}_{\pi^*}[L^{smooth}(\theta^*)|s_0]$ *represents the smoothing gap*

- $|D|$: *The support size of the initial state distribution $\mathbb{P}_{s_0}$ (equals dataset size under uniform sampling).*

- $\mathcal{C}_{\lambda,\beta}^{\pi_\theta}(s_0)$: *A concentrability coefficient measuring how well $\pi_\theta$ can reach states visited under $\pi^*$. Small $\mathcal{C}_{\lambda,\beta}$ indicates poor exploration of optimal trajectories, leading to weak gradients—the root cause of the cascade failure.*

**Remark 2** (Standard Assumptions in Policy Gradient Theory). *Following standard policy gradient analyses (Mei et al., 2020; Agarwal et al., 2021; Shen, 2025), our bounds assume the policy gradient $\|\nabla V_{\lambda,\beta}^{\pi_\theta}(D)\| \leq \epsilon$ with bounded Lipschitz constants. This is an asymptotic convergence result that holds in a neighborhood where $\min_{a,s}\pi_\theta(a|s)$ remains bounded away from zero.*

**Remark 3** (On the Smoothing Gap Term). *The smoothing gap $\Phi^{\pi_\theta,\pi^*}(s_0) := \mathbb{E}_{\pi_\theta}[L^{smooth}(\theta)|s_0] - \mathbb{E}_{\pi^*}[L^{smooth}(\theta^*)|s_0]$ involves expectations under different policies. This is not an off-policy evaluation scenario requiring importance weighting. Rather, it arises from algebraic decomposition of the*

*EPO objective $V_{\lambda,\beta}^{\pi} = V^{\pi} + \lambda H(\pi) - \beta_k \mathbb{E}_{\pi}[L^{smooth}]$. Each expectation is a well-defined analytical quantity; the comparison is valid because both contribute to bounding the overall performance gap $V^{\pi^*}(s_0) - V^{\pi_\theta}(s_0)$ through the performance difference lemma.*

**Remark 4** (Interpretation of the $|D|^2$ Scaling). *The factor $|D|^2$ in the optimization error term arises from relating the dataset-averaged gradient bound $\|\nabla V_{\lambda,\beta}^{\pi_\theta}(D)\| \leq \epsilon$ to per-query concentrability coefficients $C_\lambda^{\pi_\theta}(s_0)$. The bound can equivalently be interpreted per-query: for each $s_0 \in D$,*

$$V^{\pi^*}(s_0) - V^{\pi_\theta}(s_0) = O\left(\frac{\epsilon^2}{\lambda C_\lambda^{\pi_\theta}(s_0)}\right) + bias\ terms \tag{8}$$

*The dataset-averaged bound then follows by expectation over $s_0 \sim D$. While the bound scales with $|D|^2/C_\lambda$, in practice larger datasets typically improve $C_\lambda$ through better exploration, mitigating this dependence.*

## 7 CONCLUSION

In this work, we identified and addressed the exploration-exploitation cascade failure which is a fundamental challenge unique to training LLM agents in multi-turn environments with sparse rewards. Our proposed EPO framework addresses this through trajectory-aware entropy computation, entropy smoothing regularization, and adaptive phase-based weighting—mechanisms that together prevent the dangerous entropy oscillations that destabilize training. Empirical results demonstrate up to 152% performance improvement on ScienceWorld and 19.8% on ALFWorld, transforming previously untrainable scenarios into smoothly converging optimization problems. This work establishes that multi-turn LLM agent training requires fundamentally different entropy control than traditional RL, opening new directions for developing effective training methods for LLM Agents.

## 8 REPRODUCIBILITY STATEMENT

Our work is reproducible. The source code is publicly available at https://anonymous.4open.science/r/EPO-E637/. We provide the algorithm process of our method, Entropy-smoothed Policy Optimization (EPO), in Algorithm 1. Further implementation details, including the experimental setup and hyperparameters for all experiments, are described in Appendix B. To offer a deeper understanding of our model's behavior and the impact of its different components, we include a model study in Appendix B.4 and a pytorch-style pseudocode of EPO in Appendix C.

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

CONTENTS

# A  THEORETICAL ANALYSIS OF ENTROPY-SMOOTHED POLICY OPTIMIZATION

## A.1  PRELIMINARY SETUP

We build upon the analytical framework from the reference work (Shen, 2025; Sutton et al., 1999; Klein et al., 2023), leveraging their established lemmas without reproducing full proofs.

**Lemma 5** (Performance Difference). *For any $h \in \{0, 1, \ldots, H-1\}$ and state $s \in \mathcal{S}$, the performance difference between any two policies $\pi$ and $\pi'$ is:*

$$V_h^\pi(s) - V_h^{\pi'}(s) = \mathbb{E}_\pi \left[ \sum_{t=h}^{H-1} A_t^{\pi'}(s_t, a_t) \middle| s_h = s \right]$$

**Lemma 6** (Entropy Bias). *For the entropy-regularized objective with optimal policy $\pi_\lambda^* = \arg\max_\pi V_\lambda^\pi(\mathcal{D})$:*

$$V^{\pi^*}(s_0) - V^{\pi_\theta}(s_0) \leq V_\lambda^{\pi_\lambda^*}(s_0) - V_\lambda^{\pi_\theta}(s_0) + \lambda H \log \frac{|\mathcal{A}|}{|\mathcal{A}_H^*(s_0)|^{\frac{1}{H}}}$$

*where $\mathcal{A}_H^*(s_0)$ denote the set of optimal action sequences for a given initial state $s_0$ and a horizon of $H$.*

**Lemma 7** (Performance Bound under Gradient Norm). *If $\|\nabla V_\lambda^{\pi_\theta}(\mathcal{D})\| \leq \epsilon$:*

$$V_\lambda^{\pi_\lambda^*}(s_0) - V_\lambda^{\pi_\theta}(s_0) \leq \frac{1}{2\lambda} \frac{|\mathcal{D}|^2}{C_\lambda^{\pi_\theta}(s_0)} \epsilon^2$$

**Lemma 8** (Entropy Gradient). *For a softmax policy $\pi_\theta$:*

$$\nabla \mathcal{H}(\pi_\theta) = -\mathbb{E}_{\pi_\theta} \left[ \sum_{h=0}^{H-1} \nabla \log \pi_\theta(a_h|s_h) \sum_{t=h}^{H-1} \log \pi_\theta(a_t|s_t) \right]$$

## A.2  THE EPO PERFORMANCE BOUND

The EPO objective $V_{\lambda,\beta}^{\pi_\theta}$ can be understood as a three-fold optimization target: maximizing returns (**exploitation**), encouraging policy diversity (**exploration**), and constraining exploration behavior over time (**stabilization**). This addresses the core challenge that standard maximum entropy RL solves "whether to explore," but not "how to explore stably."

**Definition 9** (EPO Objective). *The EPO-regularized objective is:*

$$V_{\lambda,\beta}^{\pi_\theta}(\mathcal{D}) := V^{\pi_\theta}(\mathcal{D}) + \lambda \mathcal{H}(\pi_\theta) - \lambda \beta_k L^{smooth}(\theta)$$

*where $L^{smooth}(\theta)$ is defined as in Equation 6:*

$$L^{smooth}(\theta) = \frac{1}{B} \sum_{j=1}^{B} \frac{1}{T} \sum_{t=0}^{T-1} \frac{1}{|\tau_{j,t}|} \sum_{i=1}^{|\tau_{j,t}|} \mathcal{P}_{j,t,i}$$

*with penalty function:*

$$\mathcal{P}_{j,t,i} = \begin{cases} 0, & \text{if } \kappa_l \bar{H}^{\mathcal{W}_k} \leq \mathcal{H}_{j,t,i} \leq \kappa_r \bar{H}^{\mathcal{W}_k} \\ \alpha, & \text{otherwise} \end{cases}$$

**Definition 10** (Dataset Support Size). *Given dataset $D$ with initial state distribution $\mathbb{P}_{s_0}$, we define $|D| = |supp(\mathbb{P}_{s_0})|$ as the cardinality of the support of the initial state distribution.*

**Definition 11** (Concentrability Coefficient). *For query $s_0$ and policy $\pi_\theta$, the concentrability coefficient is:*

$$\mathcal{C}_\lambda^{\pi_\theta}(s_0) = C_d^2 \cdot \min_{t,s \in \mathcal{S}(s_0)} \mathbb{P}_t^{\pi_\theta}(s) \left( \min_a \pi_\theta(a|s) \right)^2 \cdot \min_{t,s \in \mathcal{S}(s_0)} \frac{\mathbb{P}_t^{\pi_\theta}(s)}{\mathbb{P}_t^{\pi_\lambda^*}(s|s_0)} \tag{9}$$

*where $C_d = (|A|H)^{-1/2}$, $\mathcal{S}(s_0)$ is the set of states reachable from $s_0$, and $\mathbb{P}_t^\pi(s)$ denotes the state visitation probability at timestep $t$ under policy $\pi$.*

The concentrability coefficient measures the distribution mismatch between the current policy $\pi_\theta$ and the optimal policy $\pi_\lambda^*$ on reachable states. A small $\mathcal{C}_\lambda^{\pi_\theta}(s_0)$ indicates that optimal trajectories have vanishingly small probability under the current policy, resulting in weak gradient signals and learning stagnation.

**Proposition 1** (Improved Performance Bound with EPO). *Assume the policy follows a softmax parameterization. If the EPO policy gradient satisfies $\|\nabla V_{\lambda,\beta}^{\pi_\theta}(\mathcal{D})\| \leq \epsilon$, then for any query $s_0$, the policy suboptimality is bounded by:*

$$V^{\pi^*}(s_0) - V^{\pi_\theta}(s_0) \leq \underbrace{\frac{|\mathcal{D}|^2}{C_{\lambda,\beta}^{\pi_\theta}(s_0)} \frac{\epsilon^2}{2\lambda}}_{\text{Optimization Error}} + \underbrace{\lambda H \log \frac{|\mathcal{A}|}{|\mathcal{A}_H^*(s_0)|^{\frac{1}{H}}}}_{\text{Standard Entropy Bias}} - \underbrace{\lambda\beta_k \Phi^{\pi_\theta,\pi^*}(s_0)}_{\text{Corrective Bias from Smoothing}}$$

*where:*

- $\Phi^{\pi_\theta,\pi^*}(s_0) := \mathbb{E}_{\pi_\theta}[L^{smooth}(\theta)|s_0] - \mathbb{E}_{\pi^*}[L^{smooth}(\theta^*)|s_0]$ *represents the smoothing gap*

- $|D|$: *The support size of the initial state distribution $\mathbb{P}_{s_0}$ (equals dataset size under uniform sampling).*

- $\mathcal{C}_{\lambda,\beta}^{\pi_\theta}(s_0)$: *A concentrability coefficient measuring how well $\pi_\theta$ can reach states visited under $\pi^*$. Small $\mathcal{C}_{\lambda,\beta}$ indicates poor exploration of optimal trajectories, leading to weak gradients—the root cause of the cascade failure.*

**Remark 2** (Standard Assumptions in Policy Gradient Theory). *Following standard policy gradient analyses (Mei et al., 2020; Agarwal et al., 2021; Shen, 2025), our bounds assume the policy gradient $\|\nabla V_{\lambda,\beta}^{\pi_\theta}(D)\| \leq \epsilon$ with bounded Lipschitz constants. This is an asymptotic convergence result that holds in a neighborhood where $\min_{a,s} \pi_\theta(a|s)$ remains bounded away from zero.*

**Remark 3** (On the Smoothing Gap Term). *The smoothing gap $\Phi^{\pi_\theta,\pi^*}(s_0) := \mathbb{E}_{\pi_\theta}[L^{smooth}(\theta)|s_0] - \mathbb{E}_{\pi^*}[L^{smooth}(\theta^*)|s_0]$ involves expectations under different policies. This is not an off-policy evaluation scenario requiring importance weighting. Rather, it arises from algebraic decomposition of the EPO objective $V_{\lambda,\beta}^\pi = V^\pi + \lambda H(\pi) - \beta_k \mathbb{E}_\pi[L^{smooth}]$. Each expectation is a well-defined analytical quantity; the comparison is valid because both contribute to bounding the overall performance gap $V^{\pi^*}(s_0) - V^{\pi_\theta}(s_0)$ through the performance difference lemma.*

**Remark 4** (Interpretation of the $|D|^2$ Scaling). *The factor $|D|^2$ in the optimization error term arises from relating the dataset-averaged gradient bound $\|\nabla V_{\lambda,\beta}^{\pi_\theta}(D)\| \leq \epsilon$ to per-query concentrability coefficients $C_\lambda^{\pi_\theta}(s_0)$. The bound can equivalently be interpreted per-query: for each $s_0 \in D$,*

$$V^{\pi^*}(s_0) - V^{\pi_\theta}(s_0) = O\left(\frac{\epsilon^2}{\lambda C_\lambda^{\pi_\theta}(s_0)}\right) + bias\ terms \tag{8}$$

*The dataset-averaged bound then follows by expectation over $s_0 \sim D$. While the bound scales with $|D|^2/C_\lambda$, in practice larger datasets typically improve $C_\lambda$ through better exploration, mitigating this dependence.*

*Proof.* Let $\pi_{\lambda,\beta}^* = \arg\max_\pi V_{\lambda,\beta}^\pi(\mathcal{D})$ be the optimal policy under EPO. By optimality:

$$V_{\lambda,\beta}^{\pi_{\lambda,\beta}^*}(s_0) \geq V_{\lambda,\beta}^{\pi^*}(s_0)$$

Expanding the EPO objectives:

$$V_{\lambda,\beta}^{\pi_{\lambda,\beta}^*}(s_0) - V_{\lambda,\beta}^{\pi_\theta}(s_0) \geq V_{\lambda,\beta}^{\pi^*}(s_0) - V_{\lambda,\beta}^{\pi_\theta}(s_0) \tag{10}$$

$$= \left(V^{\pi^*}(s_0) + \lambda\mathcal{H}(\pi^*|s_0) - \lambda\beta_k \mathbb{E}_{\pi^*}[L^{\text{smooth}}|s_0]\right)$$
$$- \left(V^{\pi_\theta}(s_0) + \lambda\mathcal{H}(\pi_\theta|s_0) - \lambda\beta_k \mathbb{E}_{\pi_\theta}[L^{\text{smooth}}|s_0]\right) \tag{11}$$

Rearranging:

$$V^{\pi^*}(s_0) - V^{\pi_\theta}(s_0) \leq V_{\lambda,\beta}^{\pi_{\lambda,\beta}^*}(s_0) - V_{\lambda,\beta}^{\pi_\theta}(s_0)$$
$$+ \lambda\left(\mathcal{H}(\pi_\theta|s_0) - \mathcal{H}(\pi^*|s_0)\right)$$
$$- \lambda\beta_k\left(\mathbb{E}_{\pi_\theta}[L^{\text{smooth}}|s_0] - \mathbb{E}_{\pi^*}[L^{\text{smooth}}|s_0]\right) \tag{12}$$

We use two fundamental bounds on policy entropy. First, the entropy of an optimal policy $\pi^*$ is bounded by the maximum possible entropy over its support set, the optimal trajectories $\mathcal{A}_H^*(s_0)$. Second, the entropy of any policy $\pi_\theta$ is bounded by the sum of the maximum possible single-step entropies over the horizon $H$.

$$
\begin{aligned}
\mathcal{H}_{\max} &= -\sum_{\tau \in \mathcal{A}_H^*(s_0)} \frac{1}{|\mathcal{A}_H^*(s_0)|} \log \frac{1}{|\mathcal{A}_H^*(s_0)|} \\
&= -|\mathcal{A}_H^*(s_0)| \left( \frac{1}{|\mathcal{A}_H^*(s_0)|} (-\log |\mathcal{A}_H^*(s_0)|) \right) \\
&= \log |\mathcal{A}_H^*(s_0)|
\end{aligned}
$$

$$
\begin{aligned}
\mathcal{H}(\pi_\theta | s_0) &= \mathcal{H}(a_0, a_1, \ldots, a_{H-1} | s_0) \\
&= \sum_{t=0}^{H-1} \mathcal{H}(a_t | s_t)
\end{aligned}
$$

We then obtained bounds showing that the entropy of the optimal policy concentrates on optimal actions:

$$
\mathcal{H}(\pi^* | s_0) \leq \log |\mathcal{A}_H^*(s_0)|
$$

and established a maximum entropy bound:

$$
\mathcal{H}(\pi_\theta | s_0) \leq H \log |\mathcal{A}|.
$$

Therefore:

$$
\mathcal{H}(\pi_\theta | s_0) - \mathcal{H}(\pi^* | s_0) \leq H \log |\mathcal{A}| - \log |\mathcal{A}_H^*(s_0)| = H \log \frac{|\mathcal{A}|}{|\mathcal{A}_H^*(s_0)|^{1/H}}
$$

This gives us:

$$
V^{\pi^*}(s_0) - V^{\pi_\theta}(s_0) \leq V_{\lambda,\beta}^{\pi_{\lambda,\beta}^*}(s_0) - V_{\lambda,\beta}^{\pi_\theta}(s_0) + \lambda H \log \frac{|\mathcal{A}|}{|\mathcal{A}_H^*(s_0)|^{1/H}} - \lambda \beta_k \Phi^{\pi_\theta, \pi^*}(s_0) \quad (13)
$$

We adapt the proof technique from Lemma 1. The key modification is that our Q-function now includes the smoothing term. Given the deterministic transitions in our LLM setting (where $s_{t+1}$ is fully determined by $(s_t, a_t)$), we have:

$$
Q_{t,\lambda,\beta}^{\pi_\theta}(s, a) = r(s, a) + V_{t+1,\lambda,\beta}^{\pi_\theta}(s_{t+1})
$$

where the value function incorporates smoothing penalties. Following the derivation in the reference (equations leading to their Lemma 4), we express the performance gap as:

$$
V_{\lambda,\beta}^{\pi_{\lambda,\beta}^*}(s_0) - V_{\lambda,\beta}^{\pi_\theta}(s_0) \leq \sum_{t=0}^{H-1} \mathbb{E}_{s \sim \mathbb{P}_t^{\pi_{\lambda,\beta}^*}(\cdot | s_0)} \left[ \lambda D_{KL}(\pi_\theta(\cdot|s) || \bar{\pi}_{\theta,\beta}(\cdot|s)) \right]
$$

where $\bar{\pi}_{\theta,\beta}(a|s) \propto \exp(Q_{t,\lambda,\beta}^{\pi_\theta}(s, a)/\lambda)$ is the soft-optimal policy.

The KL divergence can be bounded using the softmax parameterization:

$$
D_{KL}(\pi_\theta(\cdot|s) || \bar{\pi}_{\theta,\beta}(\cdot|s)) \leq \frac{1}{2\lambda^2} \left\| Q_{t,\lambda,\beta}^{\pi_\theta}(s, \cdot) - \lambda \theta_{s,\cdot} - c(s)\mathbf{1} \right\|_\infty^2
$$

where $c(s)$ is a normalizing constant and $\mathbf{1}$ is the all-ones vector.

The gradient of the EPO objective is:

$$
\nabla V_{\lambda,\beta}^{\pi_\theta}(\mathcal{D}) = \sum_{t=0}^{H-1} \sum_s \mathbb{P}_t^{\pi_\theta}(s) \frac{\partial \pi_\theta(\cdot|s)}{\partial \theta_{s,\cdot}} \left( Q_{t,\lambda,\beta}^{\pi_\theta}(s, \cdot) - \lambda \theta_{s,\cdot} \right)
$$

Let us define the soft advantage term $\bar{A}_{t,\lambda,\beta}^{\pi_\theta}(s)$ as:

$$\bar{A}_{t,\lambda,\beta}^{\pi_\theta}(s) := Q_{t,\lambda,\beta}^{\pi_\theta}(s,\cdot)/\lambda - \theta_{s,\cdot} - c(s)\mathbf{1} \tag{14}$$

where $c(s)$ is a state-dependent normalizing constant. Adapting the bounding technique from the reference, which combines the Cauchy-Schwarz inequality with properties of the softmax Jacobian, yields the following lower bound on the squared gradient norm:

$$\|\nabla V_{\lambda,\beta}^{\pi_\theta}(\mathcal{D})\|^2 \geq \lambda^2 C_{\lambda,\beta}^{\pi_\theta}(s_0) \sum_{t=0}^{H-1} \mathbb{E}_{s\sim\mathbb{P}_t^{\pi_{\lambda,\beta}^*}(\cdot|s_0)} \left\|\bar{A}_{t,\lambda,\beta}^{\pi_\theta}(s)\right\|_\infty^2 \tag{15}$$

This inequality is crucial as it connects the global gradient norm to the sum of local, per-state soft advantages.

where:

$$C_{\lambda,\beta}^{\pi_\theta}(s_0) = C_d^2 \min_{t,s\in\mathcal{S}(s_0)} \mathbb{P}_t^{\pi_\theta}(s)(\min_{s,a} \pi_\theta(a|s))^2 \min_{t,s\in\mathcal{S}(s_0)} \frac{\mathbb{P}_t^{\pi_\theta}(s)}{\mathbb{P}_t^{\pi_{\lambda,\beta}^*}(s|s_0)}$$

with $C_d = (|\mathcal{A}|^H)^{-1/2}$ and $\mathcal{S}(s_0)$ being the set of all states reachable from $s_0$.

Therefore:

$$V_{\lambda,\beta}^{\pi_{\lambda,\beta}^*}(s_0) - V_{\lambda,\beta}^{\pi_\theta}(s_0) \leq \frac{|\mathcal{D}|^2}{2\lambda C_{\lambda,\beta}^{\pi_\theta}(s_0)}\|\nabla V_{\lambda,\beta}^{\pi_\theta}(\mathcal{D})\|^2 \tag{16}$$

Substituting [Equation 16](#) into [Equation 13](#), and using the assumption $\|\nabla V_{\lambda,\beta}^{\pi_\theta}(\mathcal{D})\| \leq \epsilon$:

$$V^{\pi^*}(s_0) - V^{\pi_\theta}(s_0) \leq \frac{|\mathcal{D}|^2}{C_{\lambda,\beta}^{\pi_\theta}(s_0)}\frac{\epsilon^2}{2\lambda} + \lambda H \log \frac{|\mathcal{A}|}{|\mathcal{A}_H^*(s_0)|^{1/H}} - \lambda\beta_k \Phi^{\pi_\theta,\pi^*}(s_0)$$

$$\square$$

The bias reduction term $-\lambda\beta_k\Phi^{\pi_\theta,\pi^*}(s_0)$ effectively counteracts standard entropy bias when the optimal policy exhibits stable, low-variance entropy and the current policy experiences entropy violations. This correction becomes increasingly effective through the dynamic coefficient $\beta_k$, which strengthens as training progresses to enable smooth transition from exploration-focused early training to stability-focused convergence. Consequently, EPO achieves strictly better bounds than standard maximum entropy RL when $\beta_k\Phi^{\pi_\theta,\pi^*}(s_0) > 0$, directly addressing the large bias issue that becomes prohibitive in LLM settings.

## A.3 Core Proof Steps for Lemmas

**Lemma 5** (Performance Difference). *For any $h \in \{0, 1, \ldots, H-1\}$ and state $s \in \mathcal{S}$, the performance difference between any two policies $\pi$ and $\pi'$ is:*

$$V_h^\pi(s) - V_h^{\pi'}(s) = \mathbb{E}_\pi\left[\sum_{t=h}^{H-1} A_t^{\pi'}(s_t, a_t)\Big| s_h = s\right]$$

*Proof.*

$$V_h^\pi(s) - V_h^{\pi'}(s) = \mathbb{E}_\pi\left[\sum_{t=h}^{H-1} r(s_t, a_t)\Big| s_h = s\right] - V_h^{\pi'}(s) \tag{17}$$

$$= \mathbb{E}_\pi\left[\sum_{t=h}^{H-1} r(s_t, a_t) + \sum_{t=h}^{H-2} V_{t+1}^{\pi'}(s_{t+1}) - \sum_{t=h}^{H-2} V_{t+1}^{\pi'}(s_{t+1})\Big| s_h = s\right] - V_h^{\pi'}(s) \tag{18}$$

$$= \mathbb{E}_\pi\left[\sum_{t=h}^{H-1} Q_t^{\pi'}(s_t, a_t) - \sum_{t=h}^{H-1} V_t^{\pi'}(s_t)\Big| s_h = s\right] \tag{19}$$

$$= \mathbb{E}_\pi\left[\sum_{t=h}^{H-1} A_t^{\pi'}(s_t, a_t)\Big| s_h = s\right] \tag{20}$$

The key insight is telescoping the value functions and using $Q_t^{\pi'}(s_t, a_t) = r(s_t, a_t) + V_{t+1}^{\pi'}(s_{t+1})$.

$\square$

**Lemma 6** (Entropy Bias). *For the entropy-regularized objective with optimal policy $\pi_\lambda^* = \arg\max_\pi V_\lambda^\pi(\mathcal{D})$:*

$$V^{\pi^*}(s_0) - V^{\pi_\theta}(s_0) \le V_\lambda^{\pi_\lambda^*}(s_0) - V_\lambda^{\pi_\theta}(s_0) + \lambda H \log \frac{|\mathcal{A}|}{|\mathcal{A}_H^*(s_0)|^{\frac{1}{H}}}$$

*where $\mathcal{A}_H^*(s_0)$ denote the set of optimal action sequences for a given initial state $s_0$ and a horizon of $H$.*

By optimality of $\pi_\lambda^*$ for the entropy-regularized objective:

$$V_\lambda^{\pi_\lambda^*}(s_0) \ge V_\lambda^{\pi^*}(s_0) \tag{21}$$

$$V_\lambda^{\pi_\lambda^*}(s_0) - V_\lambda^{\pi_\theta}(s_0) \ge V^{\pi^*}(s_0) - V^{\pi_\theta}(s_0) + \lambda(\mathcal{H}(\pi^*|s_0) - \mathcal{H}(\pi_\theta|s_0)) \tag{22}$$

The key bounds on entropy:

$$\mathcal{H}(\pi^*|s_0) \le \log|\mathcal{A}_H^*(s_0)| \quad \text{(optimal policy is concentrated)} \tag{23}$$

$$\mathcal{H}(\pi_\theta|s_0) \le H \log|\mathcal{A}| \quad \text{(maximum entropy bound)} \tag{24}$$

Therefore:

$$\mathcal{H}(\pi^*|s_0) - \mathcal{H}(\pi_\theta|s_0) \ge \log|\mathcal{A}_H^*(s_0)| - H\log|\mathcal{A}| = -H\log\frac{|\mathcal{A}|}{|\mathcal{A}_H^*(s_0)|^{1/H}}$$

**Lemma 7** (Performance Bound under Gradient Norm). *If $\|\nabla V_\lambda^{\pi_\theta}(\mathcal{D})\| \le \epsilon$:*

$$V_\lambda^{\pi_\lambda^*}(s_0) - V_\lambda^{\pi_\theta}(s_0) \le \frac{1}{2\lambda}\frac{|\mathcal{D}|^2}{C_\lambda^{\pi_\theta}(s_0)}\epsilon^2$$

First, express the performance gap using KL divergence:

$$V_\lambda^{\pi_\lambda^*}(s_0) - V_\lambda^{\pi_\theta}(s_0) \le \sum_{t=0}^{H-1} \mathbb{E}_{s\sim\mathbb{P}_t^{\pi_\lambda^*}(\cdot|s_0)}\left[\lambda D_{KL}(\pi_\theta(\cdot|s)||\bar\pi_\theta(\cdot|s))\right] \tag{25}$$

where $\bar\pi_\theta(a|s) \propto \exp(Q_{t,\lambda}^{\pi_\theta}(s,a)/\lambda)$.

For softmax policies, the KL divergence bound:

$$D_{KL}(\pi_\theta(\cdot|s)||\bar\pi_\theta(\cdot|s)) \le \frac{1}{2\lambda^2}\left\|Q_{t,\lambda}^{\pi_\theta}(s,\cdot)/\lambda - \theta_{s,\cdot} - c(s)\mathbf{1}\right\|_\infty^2$$

The gradient norm lower bound (using Cauchy-Schwarz):

$$\|\nabla V_\lambda^{\pi_\theta}(\mathcal{D})\|^2 \ge C_d^2 \sum_{s\in\mathcal{S}(s_0)}\sum_{t=0}^{H-1}(\mathbb{P}_t^{\pi_\theta}(s))^2(\min_a \pi_\theta(a|s))^2 \tag{26}$$

$$\times \left\|Q_{t,\lambda}^{\pi_\theta}(s,\cdot)/\lambda - \theta_{s,\cdot} - c(s)\mathbf{1}\right\|_\infty^2 \tag{27}$$

Combining these with the constant:

$$C_\lambda^{\pi_\theta}(s_0) = C_d^2 \min_{t,s}\mathbb{P}_t^{\pi_\theta}(s)(\min_a \pi_\theta(a|s))^2 \min_{t,s}\frac{\mathbb{P}_t^{\pi_\theta}(s)}{\mathbb{P}_t^{\pi_\lambda^*}(s|s_0)}$$

**Lemma 8** (Entropy Gradient). *For a softmax policy $\pi_\theta$:*

$$\nabla\mathcal{H}(\pi_\theta) = -\mathbb{E}_{\pi_\theta}\left[\sum_{h=0}^{H-1}\nabla\log\pi_\theta(a_h|s_h)\sum_{t=h}^{H-1}\log\pi_\theta(a_t|s_t)\right]$$

Starting from the entropy definition:

$$\mathcal{H}(\pi_\theta) = - \sum_{s_0,a_0,\ldots,a_{H-1}} \mathbb{P}(s_0) \prod_{h=0}^{H-1} \pi_\theta(a_h|s_h) \sum_{t=0}^{H-1} \log \pi_\theta(a_t|s_t)$$

Taking the gradient and decomposing:

$$\nabla \mathcal{H}(\pi_\theta) = - \sum_{s_0,\ldots} \mathbb{P}(s_0) \prod_h \pi_\theta(a_h|s_h) \nabla \left( \sum_t \log \pi_\theta(a_t|s_t) \right) \tag{28}$$

$$- \sum_{s_0,\ldots} \mathbb{P}(s_0) \nabla \left( \prod_h \pi_\theta(a_h|s_h) \right) \sum_t \log \pi_\theta(a_t|s_t) \tag{29}$$

The first term vanishes because:

$$\sum_{a_0 \ldots a_{H-1}} \nabla \left( \prod_{h=0}^{H-1} \pi_\theta(a_h|s_h) \right) = \nabla(1) = 0$$

For the second term, using the product rule and the fact that $\mathbb{E}_{a \sim \pi_\theta(s)}[\nabla \log \pi_\theta(a|s)] = 0$:

$$\nabla \mathcal{H}(\pi_\theta) = -\mathbb{E}_{\pi_\theta} \left[ \sum_{h=0}^{H-1} \nabla \log \pi_\theta(a_h|s_h) \sum_{t=0}^{H-1} \log \pi_\theta(a_t|s_t) \right] \tag{30}$$

$$= -\mathbb{E}_{\pi_\theta} \left[ \sum_{h=0}^{H-1} \nabla \log \pi_\theta(a_h|s_h) \sum_{t=h}^{H-1} \log \pi_\theta(a_t|s_t) \right] \tag{31}$$

The last equality follows from the tower property of expectation, where terms with $t < h$ have zero expectation.

# B  EXPERIMENTS

## B.1  DETAILED EXPERIMENT SETUP

### B.1.1  BENCHMARK

We evaluate our method on two challenging and complementary benchmarks, ScienceWorld and ALFWorld, which test distinct yet crucial reasoning capabilities.

ScienceWorld (Wang et al., 2022) is a dynamic, text-based environment simulating a grade-school science lab, where the agent must solve open-ended scientific tasks. Success demands systematic hypothesis testing, common-sense reasoning about object properties, and a deep understanding of cause and effect. Its curriculum is divided into over 30 task types sourced from official study guides, primarily spanning: Physics, with tasks such as powering electrical circuits, testing the conductivity of materials, and manipulating states of matter; Chemistry, including identifying acids and bases or observing chemical reactions; and Life Science, which involves classifying organisms based on their traits. These tasks rigorously test an agent's abstract knowledge and procedural reasoning.

In contrast to the abstract challenges in ScienceWorld, ALFWorld (Shridhar et al., 2021) tests embodied reasoning in a visually-rich environment. It requires an agent to interpret high-level natural language instructions and decompose them into long sequences of low-level actions within simulated household settings. The benchmark is structured around seven main task categories designed to test long-horizon planning and language grounding: (1) Pick & Place, for simple object relocation (e.g., "Put a mug in the coffee maker"); (2) Pick Two & Place, for handling multiple objects; (3) Pick & Cool/Heat, requiring state changes using appliances; (4) Pick & Clean, involving interaction with

sinks; and more complex compositional tasks like (5) Look At Obj & Pick and (6) Examine In Light. Success in ALFWorld hinges on multi-step task planning, spatial awareness, and the ability to ground language in a physical context.

Together, these two benchmarks provide a comprehensive testbed for our agent's capabilities, spanning from abstract knowledge application in ScienceWorld to embodied task execution in ALFWorld.

### B.1.2    EVALUATION SETTING

We employ dual success rate metrics to capture different aspects of performance: **Succ.**$^{*}$ reports the average of maximum success rates across random seeds, while $\overline{\textbf{Succ.}}$ measures average performance after convergence, reflecting practical reliability. To calculate the average converged success rate ($\overline{Succ.}$), we first identify a convergence period where performance stabilizes.

In the ALFWorldenvironment, we observed that all methods exhibit similar convergence trends, with success rates plateauing after step 125. Therefore, we compute $\overline{Succ.}$ by averaging the success rates from step 125 onward (inclusive) across three random seeds. In contrast, the ScienceWorldenvironment exhibited more varied convergence behaviors across different random seeds, necessitating a per-run analysis. Specifically, the epoch ranges for computing the converged success rate in the ScienceWorldenvironment were determined as follows: In our comparison with GRPO, the evaluation windows for the three random seeds of the GRPO baseline were set to epochs 70-120, 90-120, and 90-120. In stark contrast, for our method (GRPO with EPO), these windows began significantly earlier, spanning epochs 60-80, 80-120, and 25-120, respectively. A similar trend was observed in the PPO comparison. The PPO baseline's convergence was identified late in training, with windows of 100-120, 105-120, and 105-120. When enhanced with EPO, the model converged much faster, with its evaluation periods set to 70-120, 60-120, and 60-120 for the three seeds. This detailed breakdown confirms that EPO consistently accelerates convergence across different algorithms and seeds.

Given the high variance inherent in RL, final performance scores alone can be misleading. We therefore present averaged curves to provide a more robust comparison and illustrate the performance evolution throughout the training process. We apply wandb's default running average (window size of 10) to smooth all training curves. This standard practice avoids visualization-specific tuning and ensures a fair comparison of the underlying learning trends. Additionally, we scale the variance by a factor of 0.8 for better visual clarity.

### B.1.3    BASELINES

We conduct comprehensive comparisons across multiple paradigms to evaluate the effectiveness of our proposed EPO methodology. These baselines are grouped into four categories, each representing a distinct approach to training large language model (LLM) agents.

**Prompting-based Approaches** This paradigm focuses on leveraging the in-context learning capabilities of LLMs without any parameter optimization.

The ReAct framework (Yao et al., 2023) synergizes reasoning and acting in language models. Its core innovation is the interleaving of textual reasoning traces with actions that interact with an external environment. Unlike prior methods that treated reasoning and acting as separate processes, ReAct allows the model to create and adjust high-level plans while grounding them in reality by gathering information from the environment. As a prompting-based method, ReAct's performance relies on the quality of the in-context examples and the inherent capabilities of the base model. Its operational scope is confined to single-pass inference, without mechanisms for parameter optimization or learning from experiences across multiple episodes to discover novel policies.

**Trajectory-based and Platform Methods** This category includes methods that rely on imitating expert trajectories and platforms designed for agent development.

SFT is a fundamental approach for adapting pre-trained language models to specific tasks by imitating expert-provided trajectories. The effectiveness of SFT is contingent upon the availability of a comprehensive dataset of high-quality expert demonstrations. The resulting agent's policy is inherently bounded by the scope of behaviors observed within this dataset, constraining its ability to explore novel strategies beyond the demonstrated examples.

AgentGym (Xi et al., 2024) is a comprehensive framework for building and evaluating generalist LLM-based agents, introducing a self-evolution method, AgentEvol. While AgentGym provides a valuable framework for evaluation, its AgentEvol method operates through a form of behavioral cloning. The evolution of the agent is thus guided by the quality and diversity of the initial trajectory data, which influences its sample efficiency in exploring the environment.

**General Reinforcement Learning Approaches** This group consists of well-established reinforcement learning algorithms that are not specifically designed for LLM agents but are widely used in the field.

PPO (Schulman et al., 2017) is a state-of-the-art on-policy reinforcement learning algorithm known for its stability and ease of implementation. It uses a clipped surrogate objective function to constrain policy updates. When applied to multi-turn, sparse-reward environments, credit assignment in standard PPO is performed based on the terminal reward signal. This structure can present challenges in distributing credit across a long sequence of actions, potentially leading to instabilities such as the "entropy oscillation" phenomenon.

GRPO (Shao et al., 2024) is a critic-free reinforcement learning algorithm. Instead of relying on a learned value function, it compares the performance of a group of trajectories generated from the same initial state. GRPO performs credit assignment at the trajectory level, evaluating the collective outcome of an entire episode. This design provides a holistic signal for policy updates, rather than turn-by-turn feedback, which is a consideration for learning complex, multi-step tasks.

**Agent RL Approaches** This category includes recent reinforcement learning methods that are specifically designed for training LLM agents.

GIGPO (Feng et al., 2025) extends GRPO by introducing a two-level hierarchical structure for advantage estimation. It refines the trajectory-level credit assignment of GRPO by introducing a micro-level analysis based on "anchor states." The utility of this mechanism is related to the frequency with which these anchor states are revisited, and learning is guided by the single reward signal provided at the conclusion of each episode.

RLVMR (Zhang et al., 2025) is a framework that integrates dense, process-level supervision into the reinforcement learning loop by rewarding verifiable, meta-reasoning behaviors. RLVMR shapes the agent's behavior by leveraging a "teacher" model (e.g., GPT-4) to annotate expert trajectories with meta-reasoning tags. The learning process is thus guided by the quality and potential biases inherent in these teacher-provided annotations.

### B.1.4 IMPLEMENTATION DETAILS

All of our experiments were conducted on a single server node equipped with eight NVIDIA H100 or A100 GPUs to ensure consistent and reproducible results. The training times varied based on the environment's complexity, the RL algorithm, and the GPU architecture. On the computationally intensive SciWorld benchmark, a full PPO training run required approximately 23 hours on H100s and 31 hours on A100s. The more efficient GRPO baseline was faster, completing in 16 hours on H100s and 20 hours on A100s. For the ALFWorld environment, the PPO baseline took 23 hours with H100s and 31 hours with A100s, while the GRPO baseline required 18 hours and 25 hours, respectively. Crucially, our proposed EPO method is designed for efficiency and introduces negligible computational overhead. The training times for our EPO-enhanced variants (PPO+EPO and GRPO+EPO) are effectively identical to their corresponding PPO and GRPO baselines under the same hardware configuration. The detailed hyperparameter configurations for each setup are presented in Table 3 and Table 2.

### B.2 PERFORMANCE COMPARISON

Figure 5 demonstrates the training dynamics and generalization performance of our EPO enhancement across two environments and algorithms. Our EPO variants exhibit superior convergence characteristics in both settings. In ScienceWorld, GRPO+EPO achieves early convergence around step 60 with higher peak rewards than baseline GRPO ( Figure 5(a)). Similarly, PPO+EPO in ALFWorld maintains more consistent reward accumulation with reduced oscillations ( Figure 5(b)). This improved stability stems from EPO's entropy regularization guiding exploration toward productive policy regions.

Table 2: Hyperparameter settings for the **SciWorld** environment.

| Hyperparameter | PPO | GRPO | PPO+EPO | GRPO+EPO |
|---|---|---|---|---|
| *General Setup* | | | | |
| Foundation Model | | Qwen2.5-7B-Instruct | | |
| Total RL Steps ($K$) | | 125 | | |
| Max Prompt Length | | 2048 | | |
| Max Response Length | | 512 | | |
| Test Frequency (steps) | | 5 | | |
| *Optimizer & LR Scheduler* | | | | |
| Optimizer | | AdamW | | |
| LR Scheduler | | Cosine ($num\_cycles = 0.5$) | | |
| Learning Rate | $3 \times 10^{-6}$ | $5 \times 10^{-6}$ | $3 \times 10^{-6}$ | $5 \times 10^{-6}$ |
| Warmup / Min LR Ratio | | 0.1 / 0.2 | | |
| *Batch Sizes & PPO-Specific Setup* | | | | |
| Mini-batch Size | 64 | 128 | 64 | 128 |
| Micro-batch Size | 8 | 16 | 8 | 16 |
| Critic Micro-batch Size | 4 | — | 4 | — |
| *EPO* | | | | |
| $\lambda$ | — | — | 0.001 | |
| $\beta_{start}$ | — | — | 2.0 | |
| $\beta_{end}$ | — | — | 1.0 | |
| $\kappa_l, \kappa_r$ | — | — | 0.0, 2.0 | |
| $\lambda_d$ | — | — | 3.0 | |
| $\lambda_k$ | — | — | 0.05 | |

Table 3: Hyperparameter settings for the **ALFWorld** environment.

| Hyperparameter | PPO | GRPO | PPO+EPO | GRPO+EPO |
|---|---|---|---|---|
| *General Setup* | | | | |
| Foundation Model | | Qwen2.5-3B-Instruct | | |
| Total RL Steps ($K$) | | 150 | | |
| Max Prompt Length | | 2048 | | |
| Max Response Length | | 512 | | |
| Test Frequency (steps) | | 5 | | |
| *Optimizer & LR Scheduler* | | | | |
| Optimizer | | AdamW | | |
| LR Scheduler | | Cosine ($num\_cycles = 0.5$) | | |
| Learning Rate | | $5 \times 10^{-6}$ | | |
| Warmup / Min LR Ratio | | 0.1 / 0.2 | | |
| *Batch Sizes & PPO-Specific Setup* | | | | |
| Mini-batch Size | 256 | — | 256 | — |
| Micro-batch Size | 32 | — | 32 | — |
| Critic Micro-batch Size | 16 | — | 16 | — |
| *EPO* | | | | |
| $\lambda$ | — | — | 0.001 | |
| $\beta_{start}$ | — | — | 2.0 | |
| $\beta_{end}$ | — | — | 1.0 | |
| $\kappa_l, \kappa_r$ | — | — | 0.0, 2.0 | |
| $\lambda_d$ | — | — | 3.0 | |
| $\lambda_k$ | — | — | 0.1 | |

For ScienceWorld, GRPO+EPO demonstrates clear advantages across both IID and OOD settings, achieving success rates exceeding 0.8 within 40 steps while baseline GRPO struggles to surpass 0.6

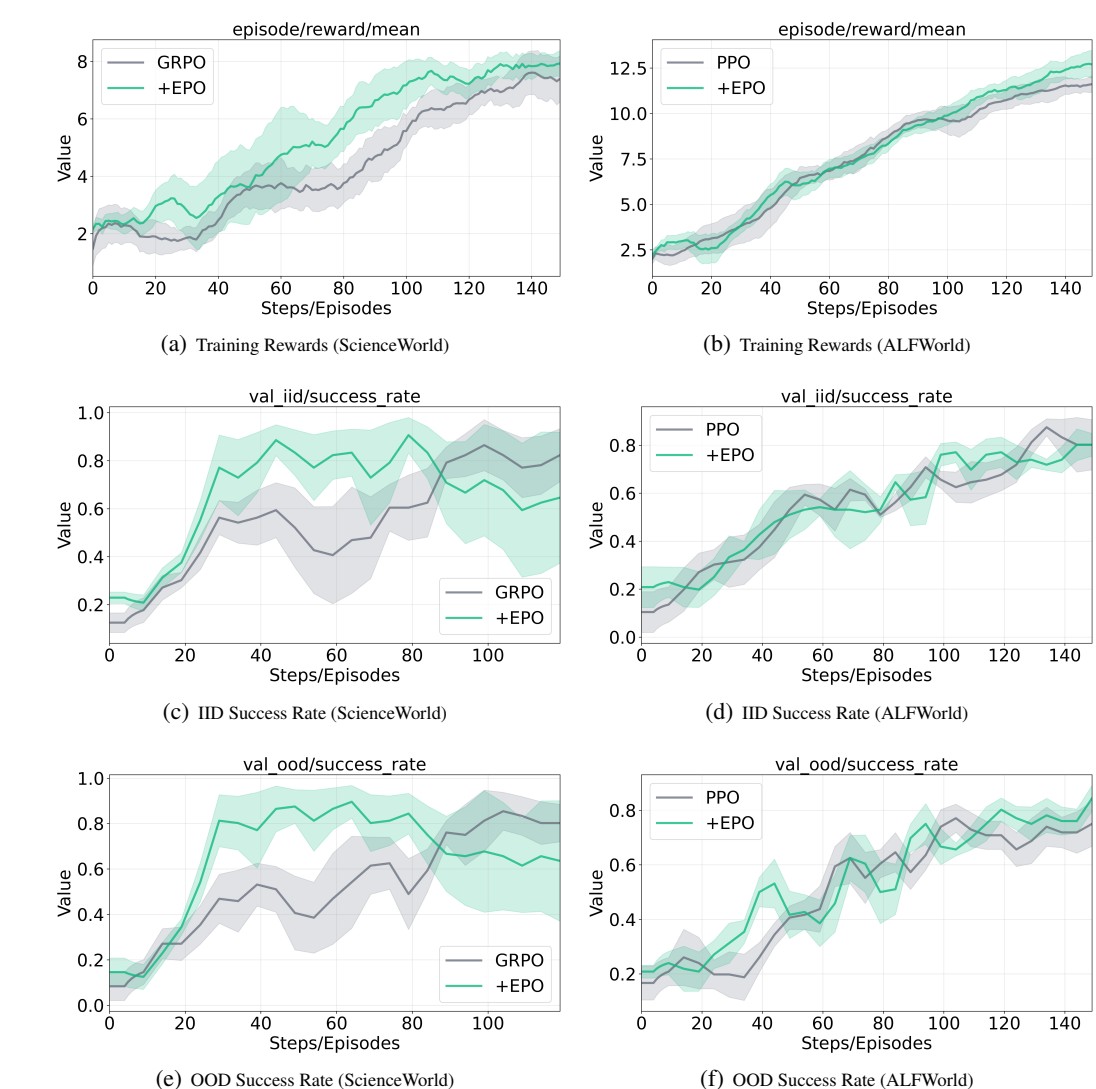

Figure 5: Training dynamics and generalization performance analysis. We present the evolution of training rewards and validation success rates across both in-distribution (IID) and out-of-distribution (OOD) evaluation settings. (a,c,e) ScienceWorld experimental results contrasting GRPO and GRPO+EPO performance across training reward accumulation, IID validation, and OOD validation metrics. (b,d,f) ALFWorld experimental results contrasting PPO and PPO+EPO under identical evaluation criteria. Our EPO enhancement exhibits significantly improved training stability and substantial performance gains across both IID and OOD evaluation scenarios against baseline methods.

(Figure 5(c),(e)). In ALFWorld, PPO+EPO prioritizes OOD robustness over IID performance. While showing comparable IID results (Figure 5(d)), PPO+EPO maintains consistent OOD success rates above 0.6 compared to baseline PPO's frequent drops below 0.4 (Figure 5(f)).

The key advantage of EPO lies in variance reduction and elimination of training oscillations. Across both environments, EPO variants show tighter confidence intervals and more reliable convergence patterns. This stability particularly benefits OOD scenarios where baseline methods exhibit substantial performance degradation. These results validate our entropy regularization approach for addressing exploration-exploitation challenges in multi-turn LLM agent training, demonstrating simultaneous improvements in generalization capability and convergence reliability.

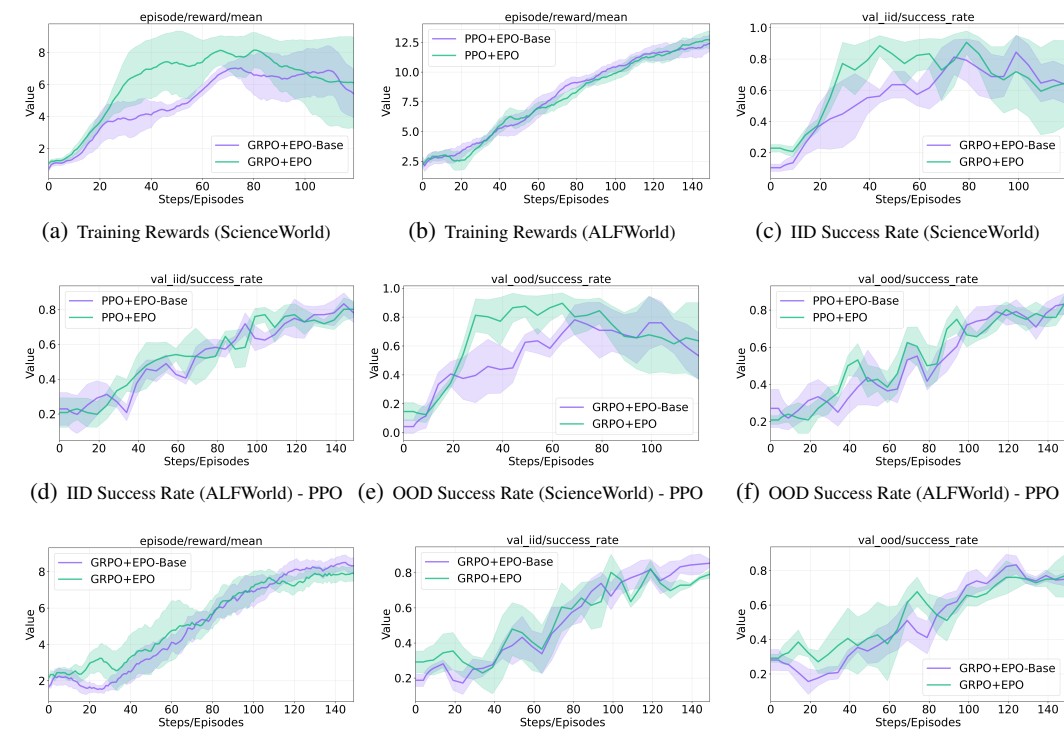

(a) Training Rewards (ScienceWorld)  (b) Training Rewards (ALFWorld)  (c) IID Success Rate (ScienceWorld)

(d) IID Success Rate (ALFWorld) - PPO  (e) OOD Success Rate (ScienceWorld) - PPO  (f) OOD Success Rate (ALFWorld) - PPO

(g) Training Rewards (ALFWorld) - GRPO  (h) IID Success Rate (ALFWorld) - GRPO  (i) OOD Success Rate (ALFWorld) - GRPO

Figure 6: Impact of the entropy smoothing regularizer on training dynamics and performance. This ablation study contrasts our full method (EPO) with a variant that excludes the entropy smoothing regularizer (EPO-base). The comparison on ScienceWorld (a,c,e) and ALFWorld (b,d,f) demonstrates that the smoothing mechanism is essential for stable RL training progression.

## B.3   ABLATION STUDY

Figure 6 extends our ablation analysis to both GRPO and PPO variants across ScienceWorld and ALF-World. In **ScienceWorld** (a,c,e), the entropy smoothing regularizer proves essential: GRPO+EPO-Base exhibits severely delayed learning with rewards remaining near 2 until step 40 and success rates plateauing at 0.6, while GRPO+EPO achieves immediate progress reaching 7-8 rewards and 0.7-0.85 success rates with a 40-50% relative improvement that persists throughout training. This pattern holds across both GRPO and PPO, confirming the mechanism's algorithm-agnostic benefits. **ALFWorld** (b,d,f) shows markedly different dynamics: both PPO variants converge to similar final performance ( 12.5 reward, 0.8 success rate), with PPO+EPO primarily demonstrating 20-episode faster convergence. This differential impact validates our theoretical framework—ScienceWorld's extreme sparsity (30+ actions before feedback) creates pathological exploration-exploitation oscil-lations that the smoothing regularizer effectively breaks by maintaining entropy within historical bounds. ALFWorld's structured feedback naturally prevents such oscillations, making smoothing beneficial for speed but not essential for convergence.

The consistent improvements across both GRPO and PPO in sparse settings confirm that entropy smoothing addresses a fundamental challenge in multi-turn optimization rather than algorithm-specific weaknesses. The adaptive $\beta_k$ weighting enables this by providing strong stabilization when oscillations are detected while relaxing constraints during natural convergence, transforming intractable sparse reward problems into smoothly converging optimization processes.

## B.4   MODEL STUDY

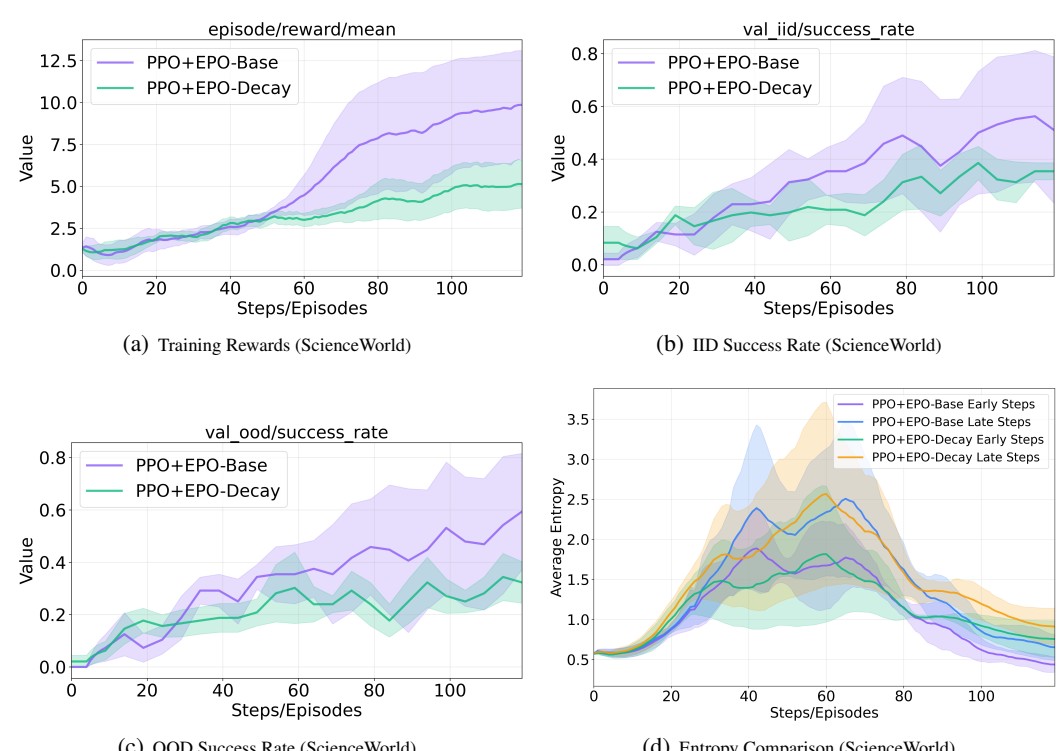

Figure 7: Performance comparison of our standard PPO+EPO-Base against PPO+EPO-Decay, which uses a decaying entropy coefficient, on the ScienceWorld benchmark. Panels (a-c) demonstrate that the dynamic decay schedule consistently degrades performance across episodic rewards and success rates. Panel (d) analyzes the intra-episode entropy for early versus late tokens, revealing that the decay schedule prematurely suppresses crucial early-turn exploration, which negatively impacts overall performance.

### B.4.1 Study of Entropy Regularization

To analyze the exploration-exploitation trade-off, we compare our standard method, **PPO+EPO-Base**, which applies a consistent entropy regularization coefficient throughout training, against an experimental variant, **PPO+EPO-Decay**. This variant was designed to test the hypothesis that a dynamic schedule could improve performance. It employs a formula to modulate the entropy weight over time: assigning a higher weight during initial training phases to promote exploration, and systematically reducing the weight in later phases to encourage exploitation.

Contrary to our hypothesis, the empirical results in Figure 7 show this strategy is counterproductive. The PPO+EPO-Decay variant consistently underperforms the baseline across all metrics, including episodic reward (a), in-distribution success rate (b), and out-of-distribution success rate (c).

Panel (d) provides insight into this failure by analyzing the intra-episode entropy, comparing the average entropy of the first 10 tokens ("Early Steps") with the last 10 tokens ("Late Steps"). While the decay schedule successfully reduces the policy's entropy in the later stages of training, it does so at a significant cost. The schedule prematurely suppresses exploration in the crucial initial turns of each episode. This insufficient early exploration locks the agent into suboptimal strategies from which it cannot recover, even as the policy becomes more deterministic. This finding underscores that for complex, multi-turn tasks, maintaining a robust and consistent exploration pressure is more effective than manually scheduling a transition towards exploitation.

### B.4.2 Study of Entropy-Shaped Advantage

We compare our Entropy-smoothed Policy Optimization (EPO) with the Entropy-based Advantage (EA) shaping method from Cheng et al. (Cheng et al., 2025b). As shown in Figure 8, while PPO+EA

improves over the baseline, our PPO+EPO is substantially superior in both final performance and convergence speed.

The primary difference lies in the gradient signal. The EA method uses a detached entropy term , which acts as an indirect intrinsic reward rather than a direct, optimizable objective. Consequently, the policy receives no gradient signal to explicitly increase its entropy. In contrast, our EPO formulation integrates entropy directly into the policy loss, enabling a direct gradient $\nabla_\theta L^H(\theta)$ to explicitly guide the policy towards more exploratory behavior. Furthermore, EA's hard clipping on the advantage bonus can induce training instability, and its myopic nature considers only instantaneous entropy. Our EPO method promotes smoother and more consistent updates by using a continuous smoothing regularizer that leverages a historical entropy window. This temporal consistency is critical for long-horizon reasoning tasks.

These theoretical advantages explain the empirical gap: PPO+EPO converges to a near-optimal success rate of almost 1.0, while PPO+EA plateaus far lower at 0.5-0.6. We posit that EA's direct advantage modification distorts the credit assignment process. In contrast, EPO's decoupled regularization preserves the integrity of the value signal, leading to more robust and effective learning.

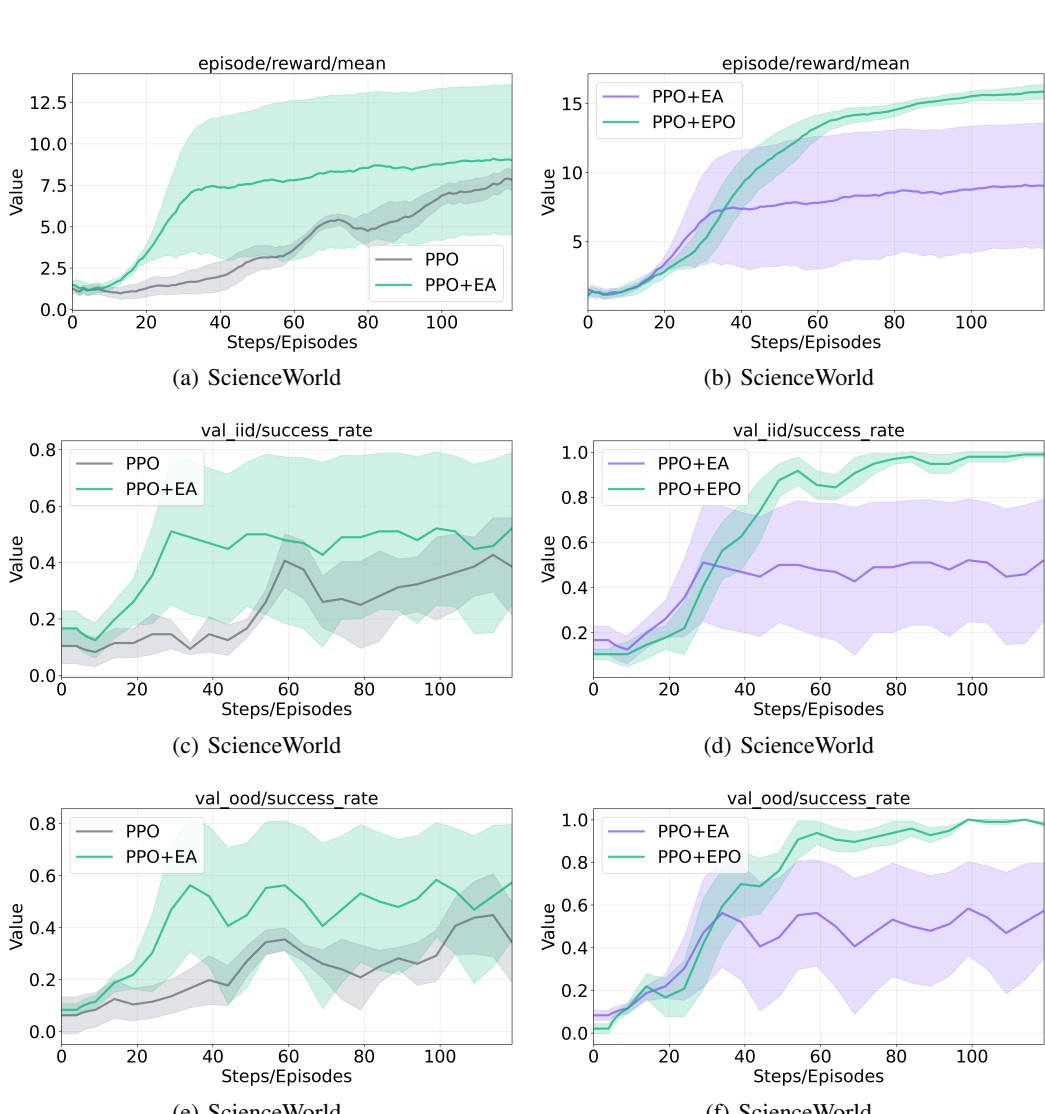

Figure 8: Performance comparison on ScienceWorld environment: vanilla PPO, PPO with Entropy-based Advantage shaping (PPO+EA) from Cheng et al. (Cheng et al., 2025b), and our PPO with Entropy-smoothed Policy Optimization (PPO+EPO). Results show episodic rewards (a,b), validation IID success rates (c,d), and OOD success rates (e,f). PPO+EPO consistently outperforms both baselines, achieving near-perfect success rates ($\sim$1.0) compared to PPO+EA's plateau at 0.5-0.6. Curves show mean values with shaded standard error across multiple seeds.

## C  CORE IMPLEMENTATION PSEUDOCODE

Algorithm 2 presents a PyTorch-style pseudocode outlining the core computational steps for our proposed Entropy-Smoothed Policy Optimization (EPO) loss. This implementation directly corresponds to the methodology described in Section 3, detailing how the base policy loss, entropy regularization, and the entropy smoothing regularizer are combined to form the final training objective for a single policy update step.

---

**Algorithm 2** PyTorch-style Pseudocode for EPO Loss Calculation

---

```
1  # Given: policy `policy_pi`, batch `data`, current epoch `k`
2  # `data` contains: old_log_probs, advantages, response_mask,
       entropy_history
3  # Hyperparameters: lambda_, kappa_l, kappa_r, alpha, K
4
5  # 1. Forward pass for current log probabilities and token-level entropy
6  logits = policy_pi(data.input_ids, data.attention_mask)
7  log_prob, entropy = get_logprob_and_entropy(logits, data.responses)
8
9  # 2. Compute the base multi-turn policy loss L^MT (e.g., PPO objective)
10 pg_loss = compute_policy_loss(
11     log_prob=log_prob,
12     old_log_prob=data.old_log_probs,
13     advantages=data.advantages,
14     response_mask=data.response_mask,
15     clip_ratio=0.2
16 )
17
18 # 3. Compute the entropy regularization loss L^H (Eq. 6)
19 entropy_loss = agg_loss(entropy, data.response_mask)
20
21 # 4. Compute the entropy smoothing regularizer
22 # 4a. Calculate historical average entropy from window W_k
23 historical_avg_entropy = data.entropy_history.mean()
24
25 # 4b. Generate token-wise penalty mask P based on historical avg (Eq. 8)
26 penalty_mask = generate_entropy_penalty(
27     current_entropy=entropy,
28     historical_avg_entropy=historical_avg_entropy,
29     min_ratio=kappa_l, max_ratio=kappa_r, penalty_weight=alpha
30 )
31
32 # 4c. Calculate smoothing loss L^smooth by aggregating penalties (Eq. 9)
33 smoothing_loss = agg_loss(penalty_mask, data.response_mask)
34
35 # 4d. Get dynamic coefficient beta_k for the current step k (Eq. 11)
36 beta_k = calculate_dynamic_beta(current_step=k, total_steps=K)
37
38 # 5. Combine entropy and smoothing terms
39 # Corresponds to [L^H(theta) - beta_k * L^smooth(theta)]
40 entropy_term = entropy_loss - beta_k * smoothing_loss
41
42 # 6. Compute the final EPO loss (Eq. 10)
43 # L^EPO = L^MT - lambda * [L^H - beta_k * L^smooth]
44 final_loss = pg_loss - lambda_ * entropy_term
45
```

---

# D SYSTEM PROMPTS

This appendix details the system prompts used to guide the language model agents in the ALFWorld and ScienceWorld environments. For each environment, we provide two versions of the prompt: one that includes historical context (previous actions and observations) and one that omits it for the initial turn. The placeholders in curly braces, such as {current_observation}, are dynamically replaced with environment-specific information at runtime.

## D.1 ALFWORLD PROMPTS

Listing 1: ALFWorld prompt (without history)

```
You are an expert agent operating in the ALFRED Embodied Environment.
Your current observation is: {current_observation}
Your admissible actions of the current situation are:
    ↪ [{admissible_actions}].

Now it's your turn to take an action.
You should first reason step-by-step about the current situation. This
    ↪ reasoning process MUST be enclosed within <think> </think> tags.
Once you've finished your reasoning, you should choose an admissible
    ↪ action for current step and present it within <action> </action>
    ↪ tags.
```

Listing 2: ALFWorld prompt (with history)

```
You are an expert agent operating in the ALFRED Embodied Environment.
    ↪ Your task is to: {task_description}
Prior to this step, you have already taken {step_count} step(s). Below
    ↪ are the most recent {history_length} observations and the
    ↪ corresponding actions you took: {action_history}
You are now at step {current_step} and your current observation is: {
    ↪ current_observation}
Your admissible actions of the current situation are:
    ↪ [{admissible_actions}].

Now it's your turn to take an action.
You should first reason step-by-step about the current situation. This
    ↪ reasoning process MUST be enclosed within <think> </think> tags.
Once you've finished your reasoning, you should choose an admissible
    ↪ action for current step and present it within <action> </action>
    ↪ tags.
```

## D.2 SCIENCEWORLD PROMPTS

Listing 3: ScienceWorld prompt (without history)

```
You are an expert agent operating in the ScienceWorld environment, which
    ↪ is a text-based virtual environment centered around accomplishing
    ↪ tasks from the elementary science curriculum.
Your current task is: {task_description}

Your current observation is: {current_observation}
Here are the actions you may take:
[
{"action": "open OBJ", "description": "open a container"},
{"action": "close OBJ", "description": "close a container"},
```

```
{"action": "activate OBJ", "description": "activate a device"},
{"action": "deactivate OBJ", "description": "deactivate a device"},
{"action": "connect OBJ to OBJ", "description": "connect electrical
    ↪ components"},
{"action": "disconnect OBJ", "description": "disconnect electrical
    ↪ components"},
{"action": "use OBJ [on OBJ]", "description": "use a device/item"},
{"action": "look around", "description": "describe the current room"},
{"action": "look at OBJ", "description": "describe an object in detail"},
{"action": "look in OBJ", "description": "describe a container's
    ↪ contents"},
{"action": "read OBJ", "description": "read a note or book"},
{"action": "move OBJ to OBJ", "description": "move an object to a
    ↪ container"},
{"action": "pick up OBJ", "description": "move an object to the
    ↪ inventory"},
{"action": "put down OBJ", "description": "drop an inventory item"},
{"action": "pour OBJ into OBJ", "description": "pour a liquid into a
    ↪ container"},
{"action": "dunk OBJ into OBJ", "description": "dunk a container into a
    ↪ liquid"},
{"action": "mix OBJ", "description": "chemically mix a container"},
{"action": "go to LOC", "description": "move to a new location"},
{"action": "eat OBJ", "description": "eat a food"},
{"action": "flush OBJ", "description": "flush a toilet"},
{"action": "focus on OBJ", "description": "signal intent on a task
    ↪ object"},
{"action": "wait", "description": "take no action for 10 iterations"},
{"action": "wait1", "description": "take no action for 1 iteration"},
{"action": "task", "description": "describe current task"},
{"action": "inventory", "description": "list your inventory"}
]

Current available actions:
{available_actions}

Now it's your turn to take an action.
You should first reason step-by-step about the current situation. This
    ↪ reasoning process MUST be enclosed within <think> </think> tags.
Once you've finished your reasoning, you should choose an appropriate
    ↪ action for the current step and present it within <action>
    ↪ </action> tags.
```

Listing 4: ScienceWorld prompt (with history)

```
You are an expert agent operating in the ScienceWorld environment, which
    ↪ is a text-based virtual environment centered around accomplishing
    ↪ tasks from the elementary science curriculum.
Your current task is: {task_description}

Prior to this step, you have already taken {step_count} step(s). Below
    ↪ are the most recent {history_length} observations and the
    ↪ corresponding actions you took: {action_history}
You are now at step {current_step} and your current observation is: {
    ↪ current_observation}
Here are the actions you may take:
[
{"action": "open OBJ", "description": "open a container"},
{"action": "close OBJ", "description": "close a container"},
{"action": "activate OBJ", "description": "activate a device"},
{"action": "deactivate OBJ", "description": "deactivate a device"},
{"action": "connect OBJ to OBJ", "description": "connect electrical
    ↪ components"},
```

```
1674    {"action": "disconnect OBJ", "description": "disconnect electrical
1675       ↪ components"},
1676    {"action": "use OBJ [on OBJ]", "description": "use a device/item"},
1677    {"action": "look around", "description": "describe the current room"},
1678    {"action": "look at OBJ", "description": "describe an object in detail"},
1679    {"action": "look in OBJ", "description": "describe a container's
        ↪ contents"},
1680    {"action": "read OBJ", "description": "read a note or book"},
1681    {"action": "move OBJ to OBJ", "description": "move an object to a
1682       ↪ container"},
1683    {"action": "pick up OBJ", "description": "move an object to the
           ↪ inventory"},
1684    {"action": "put down OBJ", "description": "drop an inventory item"},
1685    {"action": "pour OBJ into OBJ", "description": "pour a liquid into a
1686       ↪ container"},
1687    {"action": "dunk OBJ into OBJ", "description": "dunk a container into a
1688       ↪ liquid"},
1689    {"action": "mix OBJ", "description": "chemically mix a container"},
        {"action": "go to LOC", "description": "move to a new location"},
1690    {"action": "eat OBJ", "description": "eat a food"},
1691    {"action": "flush OBJ", "description": "flush a toilet"},
1692    {"action": "focus on OBJ", "description": "signal intent on a task
1693       ↪ object"},
1694    {"action": "wait", "description": "take no action for 10 iterations"},
1695    {"action": "wait1", "description": "take no action for 1 iteration"},
1696    {"action": "task", "description": "describe current task"},
        {"action": "inventory", "description": "list your inventory"}
1697    ]
1698
1699    Current available actions:
        {available_actions}
1700
1701    Now it's your turn to take an action. You should first reason
1702       ↪ step-by-step about the current situation. This reasoning process
1703       ↪ MUST be enclosed within <think> </think> tags.
1704    Once you've finished your reasoning, you should choose an appropriate
1705       ↪ action for the current step and present it within <action>
           ↪ </action> tags.
```

## E  LIMITATION AND FUTURE WORK

While EPO effectively addresses the exploration-exploitation cascade failure in multi-turn sparse-reward environments, our approach does not fully leverage memory systems (Xu et al., 2025) to enhance learning from past trajectories. Currently, EPO uses historical entropy information solely for regularization, but does not incorporate explicit memory mechanisms that could help agents recall and reuse successful behavioral patterns from previous episodes. In multi-turn settings where sparse rewards make successful trajectories particularly valuable, a memory-augmented approach could potentially accelerate learning by allowing agents to explicitly store and retrieve relevant past experiences, especially those leading to rare positive rewards.

Future work could extend EPO to vision-language model (VLM) agents operating in multi-turn visual environments, where the cascade failure may manifest differently due to the multimodal nature of observations and actions. The interplay between visual and textual entropy in VLM agents presents unique challenges—visual observations might require different entropy bounds than textual responses, and the temporal dependencies across modalities could amplify or dampen the cascade failure.

## F  USE OF LARGE LANGUAGE MODELS

We utilized Large Language Models (LLMs), such as Claude, exclusively for ancillary support in two main areas: (i) language editing and polishing of the manuscript, and (ii) coding assistance for minor

boilerplate tasks, such as generating plotting scripts and small utilities. All model-generated outputs were thoroughly reviewed, modified, and rigorously tested by the authors to ensure their accuracy and appropriateness.

The core intellectual contributions of this work—including all research ideas, algorithmic designs, experimental methodologies, data analysis, and conclusions—were conceived and validated entirely by the authors. Critically, LLMs were **not** used to generate any experimental results, create annotations or ground truth data, or influence methodological decisions. The authors assume full and sole responsibility for all content presented in this paper.

