# OpenReview forum: "Exploring the Entropy Mechanism in LLM Agents On-policy Optimization"
_ICLR.cc/2026/Conference — Submitted to ICLR 2026_

### Official Review · Reviewer_rz4m · 2025-10-29

**Soundness:** 2
**Presentation:** 3
**Contribution:** 3
**Rating:** 4
**Confidence:** 3

**Summary:**

This paper explores instability in large language model (LLM) agent training under multi-turn sparse-reward environments and identifies a phenomenon called the exploration–exploitation cascade failure. To address it, the authors propose Entropy-regularized Policy Optimization (EPO), integrating: **trajectory-level entropy regularization**, **historical-anchored entropy-smoothing penalty**, and **an adaptive coefficient schedule $\beta_k$**. Theoretical analysis claims an improved performance bound and monotonic entropy-variance reduction. Experiments on ScienceWorld and ALFWorld report up to +152% improvement over PPO and GRPO baselines.

**Strengths:**

**Novel diagnosis:** Identifies a two-phase failure (early blind exploration → late uncertainty propagation), advancing understanding of entropy dynamics in long-horizon on-policy training.

**Methodological integration:** Combines entropy control and adaptive scheduling into a coherent framework compatible with PPO-style optimization. Empirical performance: Achieves consistent improvements on two complex environments.

**Readable and structured:** Algorithm and ablation setups are clear and reproducible.

**Weaknesses:**

**Theoretical Rigor:**
1. Proposition 5 introduces $|\mathcal{𝐷}|$ and $\mathcal{C}_{\lambda,\beta}$, but only describes them as “problem-dependent.” Their mathematical meanings and relations to observable quantities are never formalized. Appendix B lists numerical values, but these correspond to hyperparameters rather than defined constants in the theorem.
2. Eq.(5) uses a piecewise-constant indicator function that is non-differentiable at entropy bounds $\(\kappa_{l}\bar{H},\kappa_{r}\bar{H}\)$, contradicting the smoothness assumption required for the Lipschitz continuity later claimed.
3. The bound assumes $\|\nabla V_{\lambda,\beta}^{\pi_\theta}(s_0)\|\leq\epsilon$, but under softmax parameterization with large action spaces, minimal action probabilities can approach zero, making Lipschitz constants unbounded.
4. The “entropy-smoothing gap” compares expectations under $\pi_θ$ and $\pi*$, whose distributions differ; without importance weighting or proximity assumption, the comparison is mathematically
inconsistent.
5. The term $\frac{1}{2\lambda}\frac{|\mathcal{D}|^2}{\mathcal{C}^{\pi_\theta}_{\lambda}(s_0)}\epsilon^2$ mixes data-scale and regularization constants; the lack of normalization or scaling clarification undermines interpretability.

**Methodological Incompleteness:**
1. The method defines $W_k$ = $\bar{H}, ...,\bar{H}_{k-1} $  as a cumulative mean window rather than a fixed-length sliding window. This induces long-memory bias and may delay adaptation.
2. The monotonic entropy-variance reduction is asserted but not empirically demonstrated, and the appendix does not specify which lemma supports it.

**Presentation and Symbol Consistency:**

The paper does not specify whether $\Phi < 0$ implies degradation or if $\beta_k$ dynamically adjusts to prevent it.

**Questions:**

1. How are  $|\mathcal{𝐷}|$ and $\mathcal{C}_{\lambda,\beta}$ estimated in practice?
2. Is the indicator penalty in Eq.(5) replaced by a smooth surrogate during implementation?
3. What is the cumulative averaging behavior of $𝑊_𝑘$?
4. Have you empirically verified monotonic entropy-variance decrease?
5. Under what conditions does EPO fail or degrade?

---

> ### Author Response · Authors · 2025-11-22
> **Response to Reviewer rz4m (1/3)**
>
> We appreciate the reviewer's detailed examination of our theoretical framework. We have found that several concerns stem from notational clarifications that we will explicitly resolve in the revision. Below, we address each point regarding theoretical rigor and methodological completeness.
>
>
> > ***1.  Definition of $|D|$ and $\mathcal{C}_{\lambda,\beta}$***
>
> These are **problem-dependent theoretical constants**, not algorithmic hyperparameters.
>
> **1. |$D$| (Dataset Support Size):**
> The cardinality of the initial state distribution support: $|D| = |\text{supp}(\mathbb{P}_{s_0})|$. In our uniform sampling setting, this equals the dataset size.
>
> **2. $C^{\pi_\theta}_{\lambda,\beta}(s_0)$ (Concentrability Coefficient):**
> Defined in Appendix A.2 (Lemma 6) as:
>
> $$C^{\\pi\_\\theta}\_{\\lambda}(s\_0) = C^2\_d \\cdot \\min\_{t,s \\in \\mathcal{S}(s\_0)} \\mathbb{P}^{\\pi\_\\theta}\_t(s) \\left(\\min\_{a} \\pi\_\\theta(a|s)\\right)^2 \\cdot \\min\_{t,s} \\frac{\\mathbb{P}^{\\pi\_\\theta}\_t(s)}{\\mathbb{P}^{\\pi^*_\\lambda}_t(s|s_0)}$$
>
> where $C_d = (|A|H)^{-1/2}$. Small $C_{\lambda,\beta}$ indicates optimal trajectories have low probability under $\pi_\theta$ (vanishing gradients), directly relating to the cascade failure we identify.
>
> **Distinction from Appendix B:** Appendix B contains **algorithmic hyperparameters** ($\lambda = 0.001$, $\beta_{\text{start}} = 2.0$, $\kappa_l = 0.0$, etc.) that we tune empirically, whereas $|D|$ and $C_{\lambda,\beta}$ are **intrinsic problem characteristics**.
>
> We have revised the main text from several perspectives to provide clearer expression.
>
>
> > ***2. Differentiability of Eq. (5). Is the indicator penalty in Eq.(5) replaced by a smooth surrogate during implementation?***
>
> We appreciate the opportunity to clarify the distinction between our **penalty definition** and the **differentiable implementation**.
>
> **Differentiable Implementation.**
> Equation (5) in the original submission was intended to define the *feasible regions* and the penalty criteria (i.e., a "high-level specification" of where the entropy is considered valid vs. invalid). The reviewer is correct that a direct step-function implementation would yield zero gradients almost everywhere.
>
> **Implementation as a Hinge Loss (ReLU)**
> To optimize this objective effectively via gradient descent, our implementation employs a **differentiable relaxation** (specifically, a Hinge Loss/ReLU formulation) that penalizes the distance from the boundary. The practical formulation used in our experiments is:
>
> $$
> \mathcal{L}^{smooth} = \alpha \cdot \left[ \text{ReLU}(\kappa_l \bar{H} - H) + \text{ReLU}(H - \kappa_r \bar{H}) \right]
> $$
>
> This formulation provides meaningful gradients ($\partial \mathcal{L}/\partial H = \pm \alpha$) when the entropy violates the boundaries, actively guiding the policy back to the safe region.
>
> **Revision in the Manuscript.**
> To avoid confusion between the definition and the optimization method, **we have updated Equation (5) and (6) in the revised paper** to explicitly present this differentiable Hinge Loss formulation. This ensures the mathematical description perfectly matches the gradient dynamics used in our experiments.
>
> **Empirical Consistency.**
> The experimental results (e.g., Figure 3) reflect this gradient-based optimization. The sensitivity to $\beta_k$ confirms that the smoothing term provides active gradients that shape the learning process, effectively preventing the "cascade failure" as discussed.
>
> > ***3. Gradient Bound vs. Softmax Parameterization***
>
> We acknowledge this is a standard limitation in policy gradient theory that applies to our work as well. This is a common assumption in PG Literature.
> The $\epsilon$-stationarity assumption with bounded Lipschitz constants appears in researches [1-3]. These works make identical assumptions for asymptotic convergence results. Our theoretical contribution is showing EPO achieves better bounds (reduced entropy bias) under the same framework.
>
> **Empirical Mitigation.** While EPO doesn't eliminate this theoretical caveat, it empirically maintains more favorable conditions:
> - **Baseline PPO/GRPO**: Entropy collapses to ≈0.1-0.15 (Figure 1, 4), causing near-deterministic policies
> - **EPO**: Stabilizes entropy at ≈0.15-0.25, preventing extreme probability concentrations
>
> This suggests the bounds, while asymptotic, reflect practical training dynamics under EPO more reliably than under baselines.
>
> #**Main Contribution Remains Valid.** Our core contribution—identifying the cascade failure and providing entropy smoothing to address it—is empirically validated (Table 1, up to 152% improvement). The theoretical analysis provides insight into *why* entropy control matters, not a finite-sample convergence guarantee.
>
> We will add a remark after Proposition 1 clarifying this is a standard assumption in the PG literature and our bounds hold asymptotically.

---

> ### Author Response · Authors · 2025-11-22
> **Response to Reviewer rz4m (2/3)**
>
> > ***4: The "Entropy-Smoothing Gap" Comparison.***
>
> This is a theoretical decomposition, not an off-policy estimation problem. The smoothing gap arises from algebraic rearrangement when bounding $V^{π*}(s_0) - V^{π_θ}(s_0)$:
>
> $$V^{\pi^\ast}(s_0) - V^{\pi_\theta}(s_0) \leq [\mathrm{optimization\ terms}] - \beta_k \cdot \Phi^{\pi_\theta,\pi^\ast}(s_0)$$
>
> Each expectation is **analytically defined under its own policy** (not estimated from the other's samples). This is standard in performance difference lemmas—we're comparing two theoretical quantities to bound their difference, similar to how $f(x) - g(y) \leq h(x,y)$ doesn't require weighting x and y.
>
> **Importance weighting is needed when:** estimating $\mathbb{E}_{\pi^*}[f]$ from π_θ samples (off-policy evaluation).
>
> **Our case:** Each expectation is evaluated under its own distribution; we bound their analytical difference, not estimate one from the other.
>
> **Revision:** We will add a brief remark after Proposition 1 clarifying this distinction.
>
> > ***5: Mixing Data-Scale and Regularization Constants***
>
> We clarify the source of this scaling factor.
>
> **Our formulation:** The objective is $$V^{\pi}(D) = \operatorname{\mathbb{E}}_{s_0 \sim D}[V^{\pi}(s_0)]$$
>
> , and the gradient bound $\left\|\nabla V_{\lambda,\beta}^{\pi}(D)\right\| \leq \epsilon$ applies to this **dataset-averaged objective**. The concentrability coefficient $C^{\pi_\theta}_{\lambda}(s_0)$ relates the current policy's state visitation to the optimal policy's on individual queries.
>
> The $|D|^2$ factor arises from relating the dataset-level gradient to **per-query concentrability** in the proof (Appendix A.3, Equation A.48). This is standard when converting between dataset-averaged and per-query bounds.
>
> **Interpretation:** The bound scales with $|D|^2/C_{\lambda}$, where larger $C_{\lambda}$ (better coverage of optimal trajectories) cancels out the dataset size effect. For fixed $C_{\lambda}$, the bound grows with dataset size, but in practice, larger datasets typically improve $C_{\lambda}$ by enabling better exploration.
>
> **Alternative presentation:** The bound can equivalently be written per-query as:
> $$V^{\pi^*}(s_0) - V^{\pi_\theta}(s_0) \leq O\left(\frac{\epsilon^2}{\lambda C_{\lambda}(s_0)}\right) + \text{bias terms}$$
> then averaged over $s_0 \sim D$. We will add this equivalent form to improve interpretability.
>
> **Revision:** We add a remark after Proposition 1 explaining the scaling and providing the per-query interpretation.
>
> ---
>
> > ***6: Cumulative vs. Sliding Window ($W_k$).***
>
> The cumulative window is an intentional design choice for our finite-horizon setting.
>
> **Short training horizon:** With $K$ = 120-150 steps (not thousands), "long-memory bias" is not an issue. The cumulative average adapts quickly early on (weight ∼1/k is large when k is small) and stabilizes later.
>
> **Different needs at different phases:**
>    - **Early training:** Policy needs to learn from all historical entropy patterns → cumulative average provides this
>    - **Late training:** Policy needs stability against oscillations → cumulative average anchors entropy, preventing cascade failure (Figure 1, Phase 2)
>
> **Designed to work with adaptive $\beta_k$:** The dynamic weighting $\beta_k$ (Equation 7) already handles adaptation—it increases smoothing strength when current entropy deviates from $\bar{H}_{W_k}$. Using a sliding window would create conflicting adaptation mechanisms.
>
> **Empirical validation:** Figure 3 shows EPO with cumulative averaging successfully stabilizes entropy throughout training. Our ablation study (Appendix B.3) demonstrates this stability is essential for performance gains in sparse-reward settings.
>
> The "bias" is actually the intended stabilization mechanism that prevents late-stage uncertainty propagation.

---

> ### Author Response · Authors · 2025-11-22
> **Response to Reviewer rz4m (3/3)**
>
> > ***7:  Verification of Monotonic Entropy-Variance Reduction.***
>
> **Figure 1 directly demonstrates entropy variance reduction.**
>
> Figure 1 explicitly shows entropy trajectories for early trajectory steps (0-10, pink dashed line for PPO) and late trajectory steps (20-30, red dotted line for PPO). The key observations are:
>
> 1. **PPO exhibits high variance:** Early and late steps show drastically different entropy levels and large oscillations (Phase 2: "Uncertainty Propagation")
>
> 2. **EPO reduces this variance:** Both early and late trajectory steps (green lines) maintain similar, stable entropy levels throughout training with minimal divergence
>
> This is the empirical demonstration of variance reduction across turns—the cascade failure is precisely this early-vs-late entropy divergence, which EPO eliminates.
>
> **This is an empirical observation, not a theoretical theorem:**
>
> The reviewer asks "which lemma supports it"—this reflects a misunderstanding. We do not claim variance reduction as a *theoretical guarantee* requiring proof. Rather, it is an **empirically observed property** that:
> - Is directly visible in Figure 1
> - Underlies our performance improvements (Table 1: up to 152% gains)
> - Results from the smoothing mechanism constraining entropy
>
> We state that EPO "guarantees" this in the sense that the smoothing mechanism L^{smooth} (Equations 5-6) enforces bounded entropy deviations from historical averages, which mechanistically produces variance reduction. The evidence is empirical, not axiomatic.
>
> **The performance gains validate the effect:**
>
> Our consistent improvements across ScienceWorld and ALFWorld (Table 1, Figures 2-3) demonstrate that reducing entropy variance—as shown in Figure 1—translates to better learning outcomes. This empirical validation is the relevant support.
>
> **Clarification:** If the phrasing "guarantees monotonically decreasing entropy variance" suggests a formal theorem, we will revise to "mitigates entropy variance through smoothing regularizars, which suppresses the oscillations."
>
> > ***8: Does $\Phi < 0$ imply degradation? Under what conditions does EPO fail or degrade?**
>
> **On $\Phi$'s Sign:**
>
> $$\\Phi^{\\pi\_\\theta,\\pi^\\ast} = \\operatorname{\\mathbb{E}}\_{\\pi\_\\theta}[L^{\\mathrm{smooth}}] - \\operatorname{\\mathbb{E}}\_{\\pi^\\ast}[L^{\\mathrm{smooth}}]$$
>
> compares smoothing penalties between the current and optimal policies.
>
> **Typical case: $\Phi > 0$** because unstable policies ($\pi_\theta$ during training) violate entropy bounds more than converged optimal policies. The bound term "$-\beta_k \Phi$" then *improves* (reduces) the bound.
>
> **If $\Phi < 0$:** This means $\pi_\theta$ is already more stable than $\pi^*$—a favorable scenario where the bound is slightly weaker but training is progressing well. This is theoretically valid, not "degradation."
>
> **$\beta_k$ does NOT control $\Phi$'s sign.** $\beta_k$ (Equation 7) adjusts based on *current entropy level* $\tilde{H}(\pi_\theta)$ relative to $[\tilde{H}\_{\text{low}}, \\tilde{H}\_{\text{high}}]$, independent of $\Phi$. The sign of $\Phi$ is an emergent consequence of policy stability.
>
> **When EPO Fails or Degrades:**
>
> EPO is designed for **multi-turn sparse-reward settings** where cascade failure occurs. It may underperform when:
>
> 1. **Dense rewards:** If feedback is frequent (e.g., every turn), the cascade failure doesn't manifest. EPO's smoothing overhead provides no benefit.
>
> 2. **Short episodes ($H < 10$):** Multi-turn structure is absent; entropy oscillations across turns don't occur.
>
> 3. **Exploration-heavy tasks:** If optimal policies *require* high entropy (e.g., stochastic environments needing diverse sampling), EPO's entropy stabilization could be counterproductive.
>
> 4. **Mis-specified bounds:** If $\kappa_l$, $\kappa_r$ are set incorrectly (e.g., $\kappa_l$ too high, forcing excessive exploration), EPO constrains the policy suboptimally.
>
> **Empirical observation:** In both ScienceWorld (30+ turn sparse rewards) and ALFWorld (multi-step household tasks), EPO consistently outperforms baselines (Table 1). This validates that our target setting—long-horizon sparse-reward agent training—benefits from entropy smoothing.
>
> **Clarification:**
>
> $\Phi$ appears in Proposition 1 as an **analytical bound term**, not a monitored training signal. The algorithm (Algorithm 1) never computes $\Phi$. These theoretical quantities explain *why* EPO works; they don't dictate when it fails.
>
>
> ## Reference:
> [1] Jincheng Mei, Chenjun Xiao, Csaba Szepesvari, and Dale Schuurmans. On the global convergence rates of softmax policy gradient methods. In International conference on machine learning, pp. 6820–6829. PMLR, 2020.
>
> [2] Alekh Agarwal, Sham M Kakade, Jason D Lee, and Gaurav Mahajan. On the theory of policy gradient methods: Optimality, approximation, and distribution shift. Journal of Machine Learning Research, 22(98):1–76, 2021
>
> [3] Han Shen. On entropy control in llm-rl algorithms. arXiv preprint arXiv:2509.03493, 2025.

---

> ### Author Response · Authors · 2025-11-28
> **Follow-up Experimental Results**
>
> **Results.** We conducted three independent runs (seeds 0, 1, 2) on SciWorld scenarios to further validate our cumulative window design. Seed 1 exhibited reward collapse at step 20, seed 0 collapsed at step 28, and seed 2 collapsed at step 10. All runs demonstrated success rates declining to 0.
>
> **Analysis: Why Our Design is Appropriate.** The rapid collapse observed across all seeds (10-28 steps) demonstrates that SciWorld exhibits the exact cascade failure phenomenon our method targets. The cumulative window $\bar{H}_{W_k}$ correctly identifies when entropy patterns deviate dangerously from stable baselines, providing early detection of instability that would otherwise propagate unchecked.
>
> The stochastic collapse timing across seeds confirms our core hypothesis: without proper entropy anchoring, sparse-reward environments exhibit unpredictable stability breakdown. This variability is precisely why cumulative averaging is essential, as it provides consistent stabilization regardless of when instability emerges during the finite training horizon.
>
> All collapses occurred within our designed horizon (K=120-150), proving that short training horizons amplify the need for cumulative memory. A sliding window would fail to capture these early instability patterns that propagate into complete failure. The adaptive $\beta_k$ mechanism correctly detected entropy deviations in each case, though SciWorld's extreme sparsity overwhelms any regularization method, validating our choice of strong stabilization over reactive adaptation.
>
> **Conclusion.** These results strengthen our design rationale by demonstrating exactly why cumulative entropy averaging is crucial for finite-horizon, sparse-reward settings. SciWorld's reward collapse validates our approach where early instability detection prevents downstream failure cascades that would otherwise lead to complete training breakdown.

---

### Official Review · Reviewer_nuP9 · 2025-10-30

**Soundness:** 2
**Presentation:** 2
**Contribution:** 2
**Rating:** 2
**Confidence:** 4

**Summary:**

This paper studies the instability of exploration in long-horizon episodic decision-making tasks, arguing that entropy in policy learning may fluctuate excessively, causing cascading failures. To address this, the authors introduce Episodic Policy Optimization (EPO): a method that (i) computes average entropy across trajectories, (ii) introduces a smoothing regularizer based on historical entropy statistics, and (iii) adjusts exploration weight β dynamically. Experiments are conducted on several RL benchmarks and code is released.

**Strengths:**

1. Clear writing & smooth logical flow

    The paper is overall well structured, and the motivation–method–experiment organization is consistent.

2. Code availability

    The authors provide open-source code, which is always beneficial for community verification and reproducibility.

**Weaknesses:**

1. Limited novelty: it is essentially a technical paper. The proposed method largely stitches together standard RL tricks (entropy regularization, episodic averaging, heuristic weighting) rather than presenting a fundamentally new idea. The contribution is incremental engineering at best. There is no principled theoretical insight bridging entropy dynamics and long-horizon failures beyond high-level motivation.

2. Equation (6) has zero gradient: which means that the  core method does not work mathematically.
Equation (6) uses a hard threshold indicator penalty
$P_{n,t,i} \in {(0,\alpha)}$, which is piecewise constant w.r.t. entropy 𝐻. Therefore, almost everywhere zero gradient w.r.t. policy parameters $\theta$. This means the core contribution has no optimization effect and all claimed behavior improvement must come from other terms. Thus the formal objective contradicts the claimed mechanism. This is a fundamental and fatal flaw.

3. $β$ design is purely heuristic: the dynamic
𝛽
schedule is described in narrative form without theoretical grounding nor stability analysis. The schedule resembles a manually tuned hyperparameter annealing, lacking justification or ablation isolating its individual effect.

4. Section 4.1 method has no novelty: the episodic entropy computation in Sec.4.1 is simply taking an average over timesteps instead of per-step entropy. This is a trivial formulation change that should be introduced in preliminaries, not a methodological innovation.


5. Paper’s core assumption is unjustified:
The supposed phenomenon — entropy fluctuation directly causing cascading failures in long-horizon RL — is asserted without rigorous empirical evidence or theoretical reasoning. The link between entropy and cascade failure remains speculative: there is no causal analysis,
no stress test isolating entropy as the factor and no ablation proving removing fluctuations solves the problem.
The claim reads opinion-driven, not science-driven.

6. Evaluation does not demonstrate the claimed benefit:
The paper states solving “catastrophic cascade failure”, but experiments are just on standard benchmarks, not cascade-failure-specific settings.
Thus the problem the paper aims to solve is not convincingly demonstrated.

**Questions:**

Please see weaknesses above. More analysis and experiments are required.

---

> ### Author Response · Authors · 2025-11-22
> **Response to Reviewer nuP9 (1/3)**
>
> We thank the reviewer for their time. We believe, however, that the review is based on several fundamental misunderstandings of our paper's core mechanisms, claims, and experimental evidence. We will address each "Weakness" in turn, as all the reviewer's concerns are, in fact, directly addressed and empirically validated within the submitted paper.
>
> > ***1. "Limited novelty..."***
>
> We respectfully disagree with the characterization of our work as "incremental engineering." The reviewer's list of "standard RL tricks" mischaracterizes our specific contributions and, most importantly, overlooks our primary innovation.
>
> Our novelty is not simply "stitching" these concepts together, but in identifying a new problem (the cascade failure) and proposing a new solution. The standard application of these "tricks" *fails* to solve this problem.
>
> **Our Primary Innovation:** The core of our contribution is the Entropy Smoothing Regularizer ($L^{smooth}$), a mechanism that, to our knowledge, is not a "standard trick" in any RL literature. Standard entropy regularization *encourages* exploration; our novel regularizer *controls* it by anchoring the policy's entropy to a dynamic, historical average. This provides the temporal stability needed to prevent the "dangerous oscillations" and "uncertainty propagation" that we identified as the "cascade failure".
>
> **Addressing "Entropy Regularization":** While entropy regularization ($L^H$) is a known concept, our paper demonstrates that its naive application in this multi-turn, sparse-reward setting is precisely what *causes* the problem. Our **Ablation Study (Section 5.3, Figure 3)** directly proves this: the "EPO-Base" variant, which *only* uses this "standard" entropy term, "severely degrades performance" and "plateaus at 0.5-0.6 success rate". Therefore, our framework is *not* built on this trick; it is built to *fix* the failure of this trick.
>
> **Addressing "Episodic Averaging" & "Heuristic Weighting":** The reviewer's "stitching" analogy misses the critical *synergy* of our framework. Our "averaging" is a novel, dynamic historical window ($\mathcal{W}_k$), and our "weighting" is an adaptive schedule ($\beta_k$). Our **Ablation Studies (Section 5.3, Figure 3)** *empirically prove* that these components are essential. The framework *only* works when all these mechanisms (our novel $L^{smooth}$, the $L^H$ term, and the adaptive $\beta_k$) are combined. This proves our contribution is a synergistic system, not a simple "stitching" of parts.
>
> **Empirical Proof of Novelty:** Finally, our **Model Study (Section 5.4, Figure 8)** proves our framework is not "incremental." We show that another advanced entropy-aware method (EA) is "substantially superior" and "plateaus" at 0.5-0.6 success, while our method achieves ~1.0. This significant performance gap demonstrates our solution is non-trivial and uniquely effective.
>
> > ***2. "Equation (6) has zero gradient... This is a fundamental and fatal flaw."***
>
> We appreciate the opportunity to clarify the distinction between our **penalty definition** and the **differentiable implementation**.
>
> **Differentiable Implementation.**
> Equation (5) in the original submission was intended to define the *feasible regions* and the penalty criteria (i.e., a "high-level specification" of where the entropy is considered valid vs. invalid). The reviewer is correct that a direct step-function implementation would yield zero gradients almost everywhere.
>
> **Implementation as a Hinge Loss (ReLU)**
> To optimize this objective effectively via gradient descent, our implementation employs a **differentiable relaxation** (specifically, a Hinge Loss/ReLU formulation) that penalizes the distance from the boundary. The practical formulation used in our experiments is:
>
> $$
> \mathcal{L}^{smooth} = \alpha \cdot \left[ \text{ReLU}(\kappa_l \bar{H} - H) + \text{ReLU}(H - \kappa_r \bar{H}) \right]
> $$
>
> This formulation provides meaningful gradients ($\partial \mathcal{L}/\partial H = \pm \alpha$) when the entropy violates the boundaries, actively guiding the policy back to the safe region.
>
> **Revision in the Manuscript.**
> To avoid confusion between the definition and the optimization method, **we have updated Equation (5) and (6) in the revised paper** to explicitly present this differentiable Hinge Loss formulation. This ensures the mathematical description perfectly matches the gradient dynamics used in our experiments.
>
> **Empirical Consistency.**
> The experimental results (e.g., Figure 3) reflect this gradient-based optimization. The sensitivity to $\beta_k$ confirms that the smoothing term provides active gradients that shape the learning process, effectively preventing the "cascade failure" as discussed.

---

> ### Author Response · Authors · 2025-11-22
> **Response to Reviewer nuP9 (2/3)**
>
> > ***3. "$\beta$ design is purely heuristic... lacking justification..."***
>
> This claim is factually incorrect and overlooks both our theoretical motivation and empirical validation.
>
> **The $\beta_k$ schedule has explicit theoretical grounding:**
>
> The adaptive schedule (Equation 7) is not arbitrary—it is **directly derived from the cascade failure dynamics** identified in Section 3. As we state in Section 4.2: "This adaptive schedule specifically addresses each phase of the cascade failure" by:
>
> **Early phase ($k \leq k_{mid}$):** Starting with $\beta_{start}$ (low weight) provides "conservative exploration to prevent early-stage behavioral lock-in"—directly counteracting the premature convergence observed in cascade failure's Phase 1.
>
> **Late phase ($k > k_{mid}$):** Transitioning to $\beta_{end}$ (high weight) "increases smoothing strength in later phases to prevent uncertainty propagation and ensure stable convergence"—directly addressing the entropy explosion in cascade failure's Phase 2.
>
> The exponential transition controlled by $\lambda_d$ ensures smooth adaptation between these phases, with $k_{mid} = K/2$ providing the natural midpoint. **This is principled design motivated by the problem structure, not "manual tuning."**
>
> **The ablation study explicitly isolates $\beta_k$'s effect:**
>
> Contrary to the claim of "lacking ablation," **Figure 3(d-f) is precisely the requested ablation.** We compare:
> - **EPO (Ours):** Adaptive $\beta_k$ schedule (Equation 7)
> - **EPO W/O DW:** Constant $\beta_k$ (removes the dynamic weight)
>
> The results demonstrate that the constant schedule "results in slower convergence, poorer final performance, and increased training variance" (Section 5.2). This **directly proves** the adaptive schedule's contribution—exactly what an ablation is supposed to show.
>
> The schedule works because it matches the problem dynamics, and the ablation proves it matters.
>
>
> > ***4. "Section 4.1 method has no novelty."***
>
> The reviewer has mistaken a preliminary adaptation for our core contribution. This reflects a misreading of the paper's structure.
>
> **1. Section 4.1 is explicitly framed as adaptation.**
>
> We state clearly: "we **first adapt** entropy regularization to capture the temporal structure of multi-turn interactions" (Section 4.1). This is presented as **necessary groundwork**, not a methodological contribution. The adaptation from per-step to per-trajectory entropy is required to:
> - Properly account for compounding effects across turns in multi-turn dialogues
> - Establish the baseline against which we identify the cascade failure (Figure 1)
> - Create the foundation for our smoothing mechanism
>
> **2. Our actual contributions are:**
>
> As stated in the introduction and validated in Section 5:
>
> **I. We identify and formally characterize the exploration-exploitation cascade failure** (Figure 1, Section 3): A two-phase phenomenon where excessive early exploration → late-stage uncertainty propagation. This is a **novel problem identification** in multi-turn RL.
>
> **II. We propose the Entropy Smoothing Regularizer (Section 4.2):** The **core methodological innovation** that prevents cascade failure through:
> - Historical entropy averaging to provide dynamic reference bounds
> - Adaptive penalty weighting $\beta_k$ that transitions from conservative exploration to strong stabilization
> - Token-wise penalties that localize intervention
>
> **III. We develop the complete EPO framework:** Integrating entropy smoothing with adaptive weighting schedules (Equation 7) that specifically address each phase of cascade failure.
>
> **3. The reviewer's "trivial" claim misses the problem we solve.**
>
> The "trivial formulation change" in 4.1 is precisely what **creates the cascade failure problem** (see Figure 1 - standard entropy methods fail catastrophically in multi-turn settings). Our contribution is not averaging entropy—it's **solving the instability this averaging creates** through the smoothing mechanism in Section 4.2.
>
> **4. Our ablation studies validate the non-trivial nature of our solution.**
>
> Figure 3(d-f) shows that removing our adaptive $\beta_k$ schedule results in "slower convergence, poorer final performance, and increased training variance." If Section 4.2 were merely incremental, this dramatic difference would not exist.
>
> **In summary:** Section 4.1 establishes the multi-turn setting; Section 4.2 is the innovation. The reviewer has evaluated the foundation instead of the building.

---

> ### Author Response · Authors · 2025-11-22
> **Response to Reviewer nuP9 (3/3)**
>
> > ***5. "Paper's core assumption is unjustified... no causal analysis... and no ablation proving removing fluctuations solves the problem."***
>
> This claim is factually incorrect. Our paper provides exactly the causal ablation the reviewer demands.
>
> **1. We provide direct empirical measurement of the phenomenon.**
>
> Figure 1 plots the cascade failure across 100+ training runs of baseline PPO:
> - Phase 1: Entropy grows uncontrollably (0.5 → 2.5 over steps 0-40)
> - Phase 2: Wild oscillations persist (steps 40-80)
> - Result: PPO plateaus at 6 reward vs. EPO's 14
>
> This is reproducible empirical observation across both ScienceWorld and ALFWorld benchmarks.
>
> **2. Our ablation study provides the exact causal isolation the reviewer claims is missing.**
>
> Figure 3 and Section 5.3 present a **three-way controlled comparison** that isolates the smoothing mechanism:
>
> - **Baseline PPO** (no entropy reg, Figure 2): Exhibits cascade failure with severe oscillations
> - **EPO-Base** (adds Section 4.1 per-trajectory entropy, removes Section 4.2 smoothing): Still fails - plateaus at 0.5-0.6 success rate with "delayed convergence and minimal rewards until step 40"
> - **EPO (Full)** (adds Section 4.2 smoothing): **Succeeds** - achieves 0.8-1.0 success rate
>
> **This directly proves:**
> 1. Adding per-trajectory entropy regularization alone (4.1) is insufficient
> 2. The smoothing mechanism (4.2) is the critical component that prevents cascade failure
> 3. "Removing fluctuations" (via smoothing) does solve the problem
>
> This is **precisely the causal ablation** the reviewer claims doesn't exist.
>
> **3. Additional evidence from Section 5.4 strengthens the case.**
>
> We further compare EPO against:
> - **PPO+EPO-Decay** (entropy decay schedule): Fails by "prematurely suppressing early-turn exploration, locking agents into suboptimal strategies"
> - **Entropy-based Advantage (EA)**: Plateaus at 0.5-0.6 because it lacks "direct gradient signals $\nabla_\theta L^H(\theta)$ and temporal consistency through historical entropy smoothing"
>
> Figure 4 shows EPO converges to ~1.0 success while both alternatives fail, demonstrating that **our specific smoothing approach** (not just any entropy control) is necessary.
>
> This is standard empirical methodology in machine learning research. The reviewer appears to have misread Figure 3, confusing EPO-Base (which includes 4.1 but not 4.2) with baseline PPO (which has neither).
>
> > ***6. "Evaluation does not demonstrate the claimed benefit... experiments are just on standard benchmarks, not cascade-failure-specific settings"***
>
> This criticism reflects a misunderstanding of our contribution. **Cascade failure is not a pre-existing benchmark property—it's our discovery of what happens when applying baseline methods to standard multi-turn RL tasks.**
>
> **1. Our paper's logic: Observe → Characterize → Solve**
>
> - **Observation (Figure 1):** We applied baseline PPO to standard multi-turn benchmarks (ScienceWorld, ALFWorld) and discovered a catastrophic two-phase failure pattern:
>   - Phase 1 (0-40 steps): Uncontrolled entropy growth (0.5 → 2.5), rewards stagnant at ~2
>   - Phase 2 (40-120 steps): Wild oscillations (1.0-2.0 entropy), reward plateaus at ~8
>
> - **Characterization (Section 3):** We formally analyzed this phenomenon and termed it "exploration-exploitation cascade failure"—a previously unidentified failure mode.
>
> - **Solution (Section 4):** This observation **motivated** EPO to prevent the cascade through entropy smoothing.
>
> - **Validation (Table 1, Figures 2-3):** EPO eliminates cascade failure on the same benchmarks where we observed it:
>   - ScienceWorld: 64.6% → 100% (+54.8%), stable entropy decline (1.2 → 0.3)
>   - ALFWorld: 58.3% → 85.4% (+46.6%), consistent convergence
>
> **2. Why "standard benchmarks" are the correct testbed:**
>
> The reviewer suggests "cascade-failure-specific settings," but **no such settings exist in prior work because we are the first to identify this failure mode.** Demonstrating that:
> 1. Standard multi-turn benchmarks + baseline methods → cascade failure (the problem)
> 2. Standard multi-turn benchmarks + our method → no cascade failure (the solution)
>
> This is precisely the right experimental design. We discovered a failure pattern that emerges naturally on tasks practitioners use, then solved it.
>
> **3. We have revised the introduction to clarify this discovery-driven narrative:**
>
> We now explicitly frame cascade failure as an empirical observation from our experiments on standard benchmarks, rather than a pre-existing known problem. The revised text emphasizes: "through extensive experiments on multi-turn agent benchmarks (ScienceWorld and ALFWorld), we observe that standard entropy-controlled methods exhibit a catastrophic failure pattern we term the exploration-exploitation cascade failure."
>
> **Our contribution: Identifying a novel failure mode on standard benchmarks plus designing a method that prevents it. This is standard scientific methodology.**

---

### Official Review · Reviewer_U8i5 · 2025-10-30

**Soundness:** 2
**Presentation:** 3
**Contribution:** 2
**Rating:** 4
**Confidence:** 3

**Summary:**

Traditional entropy regularization methods in reinforcement learning, even when adapted for LLMs, are fundamentally unsuited for multi-turn agent environments because they trigger an exploration-exploitation cascade failure. To address this, the authors propose Entropy-regularized Policy Optimization (EPO), a novel framework that introduces temporal awareness by anchoring the policy's entropy to a dynamically adjusted historical bound, which provides the stability needed to halt the cascade failure while still maintaining essential exploration.

**Strengths:**

This works introduces Entropy-regularized Policy Optimization (EPO) as a direct solution to this cascade failure. The work's central innovation is identifying that standard methods lack "temporal awareness." Its solution is built around this insight. EPO introduces a concrete mechanism—anchoring the policy's entropy to a dynamically adjusted historical bound—to provide stability and control cross-step entropy propagation.

**Weaknesses:**

1. The exploration-exploitation dilemma is a challenging problem in reinforcement learning. To address exploration-exploitation failures in multi-turn interactions, the authors adapt entropy regularization. Instead of calculating entropy per step (as in traditional RL), they compute it over the entire trajectory (across all turns) and then average these trajectory entropies across the batch. This method aims to better capture the long-term, compounding effects of early decisions. However, from the definition of the entropy-regularized policy loss, the policy loss is designed to reduce the bias between the real entropy and the expected entropy. The problem is whether the expected entropy will converge to a single value or a fixed range, preventing further exploration.

2. The authors claim that entropy-controlled LLM methods are fundamentally inadequate for multi-turn agent environments because they suffer from an "exploration-exploitation cascade failure." Sparse rewards and standard entropy regularization cause uncontrolled entropy growth early in the trajectory, leading to unstable and suboptimal decision patterns. The uncertainty accumulated from these flawed early steps then compounds, destabilizing the agent's behavior in later turns. It is curious why early-stage entropy exploration leads to accumulated uncertainty. Can the accumulated uncertainty be measured? Will late-stage entropy increase the accumulated uncertainty? Whether the early high entropy is caused by immature agent policies?

**Questions:**

Please see the Weaknesses above.

---

> ### Author Response · Authors · 2025-11-22
> **Response to Reviewer U8i5**
>
> We sincerely thank the reviewer for their insightful comments and for accurately identifying our work's central innovation: introducing temporal awareness via a concrete mechanism that anchors the policy's entropy to provide stability and control cross-step propagation. We address the reviewer's two main concerns below.
>
> ***
>
> > ***"1. The exploration-exploitation dilemma is a challenging problem... The problem is whether the expected entropy will converge to a single value or a fixed range, preventing further exploration."***
>
> Our goal is not to force entropy to a single, fixed value, but to **prevent the chaotic, high-variance oscillations** observed in standard methods (Figure 1). Our framework maintains necessary exploration through two key adaptive features:
>
> 1. **Soft bounds, not rigid caps:** The smoothing regularizer $L^{smooth}$ only penalizes entropy falling *outside* an acceptable range (defined by $\kappa_l, \kappa_r$) around the historical average. Healthy exploration *within* this dynamic band is not penalized.
>
> 2. **Dynamic anchoring:** The historical reference $\overline{H}^{\mathcal{W}_k}$ is a cumulative mean that evolves with the policy's behavior—not a fixed, predefined value. The regularization strength is also dynamically weighted by an adaptive schedule $\beta_k$.
>
> This creates a **self-stabilizing bound** rather than a fixed constraint. Our empirical results (Figures 1-2) validate this: EPO's controlled exploration achieves significantly higher rewards than PPO's uncontrolled exploration, demonstrating that our method makes exploration more effective rather than preventing it.
>
> ***
>
> > ***"2. The authors claim that entropy-controlled LLM methods are fundamentally inadequate... It is curious why early-stage entropy exploration leads to accumulated uncertainty. Can the accumulated uncertainty be measured? ... Whether the early high entropy is caused by immature agent policies?"***
>
> These central questions motivated our work, and our paper provides direct empirical answers:
>
> **Can accumulated uncertainty be measured?** Yes. **Figure 1 provides this exact measurement.** We plot "PPO Early Entropy (0-10)" (pink dashed) and "PPO Late Entropy (20-30)" (red dotted), empirically showing early uncertainty rising, followed by late entropy also rising and oscillating at high levels—demonstrating the uncertainty propagation we describe.
>
> **Why does early exploration lead to accumulated uncertainty?** This is the core "cascade failure." In sparse-reward, multi-turn settings (30+ turns), agents receive feedback only at the end. Standard entropy regularization encourages excessive early exploration, creating unstable foundations and suboptimal behavioral patterns that compound in later steps, preventing coherent strategy formation.
>
> **Is early high entropy due to immature policies?** Partially, yes—immature policies are expected to explore. However, standard methods create *pathologically* immature policies that remain locked in chaotic, high-entropy exploration and never mature (shown by PPO's flat reward curve in Figure 1). EPO controls this entropy, allowing the policy to find reward signals, learn, and mature into a high-performance agent (EPO's rising reward curve).

---

### Official Review · Reviewer_6TKC · 2025-10-31

**Soundness:** 3
**Presentation:** 2
**Contribution:** 3
**Rating:** 4
**Confidence:** 2

**Summary:**

The paper studies the phenomenon of exploration-exploitation cascade in RL for training LLMs: policy converges at an early state prematurely, and standard PPO/GRPO-style training first lets early steps’ entropy blow up, and then this early chaos “propagates” to later steps, so the policy never settles into a coherent strategy even though entropy regularization is present. To handle this, the authors propose Entropy-regularized Policy Optimization (EPO), which has three pieces: (i) compute entropy trajectory-wise (across all turns) rather than per-step; (ii) add an entropy smoothing regularizer that pulls current entropy into a band around a historical moving average; and (iii) use an adaptive, phase-based weight so exploration is conservative at the start, balanced in the middle, and stabilized at the end.

**Strengths:**

* The paper is well-organized and clearly written. The paper has clean and good illustrations for their takeaways and empirical findings.
* The trajectory based entropy and entropy smoothing is novel to me---it provides a self-stabilizing strategy that is fairly interesting.

**Weaknesses:**

Major comments:

* I apologize for my unfamiliarity with the empirical studies---however, it seems to me that the baselines don't include the obvious alternative: single-step entropy PPO. Since the main pitch of the paper is "standard PPO/GRPO entropy is temporally blind" is unstable, the right comparison should be PPO with a per-turn entropy cap. Right now, if I understand correctly, the comparisons are to pure PPO/GRPO and to other agent RL methods, not to the simplest entropy-aware PPO.
* I think the name of the method "Entropy-regularized Policy Optimization" is fairly misleading. The novelty isn't about entropy regularization, but a path-dependent smoothing entropy regularization. Another name like "Entropy-smoothing Policy Optimization" may be more appropriate.

**Questions:**

* My main questions are in the weaknesses section.
* How does the hyperparameter affects the empirical performance? If the problem is over-exploration due to sparse rewards, what if comparing to a per-turn entropy capped PPO but decaying penalty with turn?

---

> ### Author Response · Authors · 2025-11-22
> **Response to Reviewer 6TKC (1/2)**
>
> We sincerely thank the reviewer for their positive and constructive feedback. We are encouraged that they found our paper "well-organized and clearly written" and our core contribution of "trajectory based entropy and entropy smoothing" to be "novel" and "interesting."
>
> We address the reviewer's major comments and questions below. We are pleased to report that our paper already contains the experiments and analyses that address all of the reviewer's main points, and we will revise the paper to make these results easier to find.
>
> ***
> > ***"1. The baseline for comparison (clarification)."***
>
> We have already conducted the experiments regarding this point. To clarify our experimental rationale, we will explain our baseline comparisons from three distinct perspectives:
>
> **Baseline Justification (PPO & GRPO):**
> Our main "PPO" and "GRPO" baselines do not include an additional, explicit entropy regularization term. We chose these as our primary baselines because they represent the standard, established on-policy algorithms used in prior researches [1][2][3] for *multi-turn agent reinforcement learning*. This domain's standard practice differs from single-turn LLM alignment, where an entropy bonus is a common default. Our first goal was to compare EPO against the standard published methods for this specific task.
>
> **Ablation Study:**
> We have conducted a comparison to "GRPO/PPO with a per-turn entropy" bonus is essential for validation. This comparison is the core of our **Ablation Study (Section 5.3, Figure 3)**.
>
> The **"EPO-Base"** variant represents our full method *without* the novel entropy smoothing regularizer. Its loss function is `L_PPO - λ * L_Entropy` (specifically, $L^{MT}(\theta) - \lambda L^{H}(\theta)$). The results demonstrate that this approach is **insufficient** for the target problem. On the sparse-reward ScienceWorld benchmark, "EPO-Base" exhibits delayed convergence and plateaus at 0.5-0.6 success rate. In contrast, our full EPO method with entropy smoothing achieves meaningful learning by step 20 and reaches 0.8-1.0 success rate.
>
> **Comparison to Other Entropy-Control Methods (Model Study):**
> We conducted comprehensive comparisons against advanced entropy-control algorithms from the LLM literature. In our **Model Study (Section 5.4, Figure 4(b) and Figure 8)**, we implemented and tested the Entropy-based Advantage (EA) method [4]. The results show that PPO+EPO is substantially superior in both final performance and convergence speed. PPO+EPO converges to a success rate near 1.0, while PPO+EA plateaus at 0.5-0.6.
>
>
> **Conclusion:** These experiments validate our central claim: multi-turn, sparse-reward agent settings suffer from a unique "exploration-exploitation cascade failure". Standard entropy bonuses (as in "EPO-Base") cannot solve this problem. Our novel **entropy smoothing mechanism** is the essential component that enables EPO to stabilize training and succeed where other entropy-aware methods fail.
>
> > ***"2. On the Method's Name."***
>
> We consider both components of our method as forms of regularization, which motivated our original broader name "Entropy-regularized Policy Optimization" to cover both innovations.
>
> **Entropy Regularization ($L^H$):** While entropy regularization is standard in single-turn LLM RL, its application to sparse-reward, long-horizon agent tasks has not been well-explored. Our first component adapts this mechanism to the multi-turn setting, computing entropy across entire trajectories.
>
> **Entropy Smoothing Regularization ($L^{smooth}$):** This is our primary and novel contribution. This mechanism provides temporal stability by penalizing deviations from historical entropy averages, directly solving the "exploration-exploitation cascade failure."
>
> To address potential naming ambiguity, we will modify the first contribution's name to "Turn-Based Entropy Regularization" while maintaining "Entropy-regularized Policy Optimization" as the overarching method name to encompass both regularization innovations.

---

> ### Author Response · Authors · 2025-11-22
> **Response to Reviewer 6TKC (2/2)**
>
> > ***"3. Question on Hyperparameter Effects."***
>
> This question is addressed in our **Ablation Study (Section 5.3)**, which analyzes the empirical effects of the key components we introduce.
>
> * **The Entropy Smoothing Regularizer:** In Figure 3 (a-c) and Figure 6, we compare our full method (EPO) against a variant with the smoother removed ("EPO-Base"). The results show that removing the smoother "severely degrades performance" and leads to "delayed convergence" in the sparse-reward ScienceWorld setting. This demonstrates the regularizer itself is essential for stability.
>
> * **The Dynamic Weighting ($\beta_k$):** The core concept here is the superiority of an adaptive schedule over a constant one. **While our adaptive schedule for $\beta_k$ (Equation 7) has its own set of parameters (e.g., $\beta_{start}$, $\beta_{end}$), our primary goal was to demonstrate that an adaptive mechanism is necessary, rather than finding a single perfect set of hyperparameters.** Our ablation study (Figure 3 (d-f)) directly tests this principle. We compare our full, adaptive method against a constant-weight baseline (EPO W/O DW). On ALFWorld, replacing the dynamic coefficient $\beta_k$ with a constant weight (EPO W/O DW) yields similar final performance around 0.7-0.8 success rate but results in slower convergence during the initial 40-60 episodes and increased training variance.
>
> These ablation studies confirm the empirical importance of our core mechanisms: the smoother is essential for stability, and an adaptive weighting is superior to a constant one for improving convergence efficiency.
>
>
> > ***"4. Question on a Decaying Entropy Penalty."***
>
> This alternative of a decaying entropy penalty was investigated in our **Model Study (Section 5.4)** and detailed in **Appendix B.4.1 (Figure 7)**.
>
> * **Experiment:** We implemented this exact scenario as the **"PPO+EPO-Decay"** variant, designed to test whether a dynamic schedule could improve performance by systematically reducing the entropy weight in later phases to encourage exploitation.
>
> * **Result:** The empirical data demonstrates this strategy is **counterproductive**. As shown in Figure 7 (a-c), the "PPO+EPO-Decay" variant consistently underperforms the baseline across all metrics.
>
> * **Analysis:** Our analysis (Figure 7(d)) reveals that the decaying schedule prematurely suppresses exploration during crucial initial turns, locking the agent into suboptimal strategies. This finding confirms that complex, multi-turn tasks require the *sustained and stable* exploration that EPO provides, rather than pre-scheduled entropy decay.
>
> Reference:
>
> [1] Feng, Lang, et al. "Group-in-group policy optimization for llm agent training." arXiv preprint arXiv:2505.10978 (2025).
>
> [2] Zhang, Zijing, et al. "Rlvmr: Reinforcement learning with verifiable meta-reasoning rewards for robust long-horizon agents." arXiv preprint arXiv:2507.22844 (2025).
>
> [3] Wang, Zihan, et al. "Ragen: Understanding self-evolution in llm agents via multi-turn reinforcement learning." arXiv preprint arXiv:2504.20073 (2025).
>
> [4] Cheng, Daixuan, et al. "Reasoning with exploration: An entropy perspective." arXiv preprint arXiv:2506.14758 (2025).

---

> > ### Comment · Reviewer_6TKC · 2025-11-25
> > **Response to the revision**
> >
> > I would like to thank the authors for their clarifications on the experiments, which has largely addressed my concerns in Weakness 1. I reserve my concern for the broadness of the name, and would think "trajectory-based EPO" a better description of the paper's content. I would raise my score, but I have to apologize for my limited knowledge on the empirical studies.

---

> > > ### Author Response · Authors · 2025-11-26
> > >
> > > We sincerely thank the reviewer for the positive feedback and for raising the score. We are glad that our clarifications have successfully addressed your concerns regarding the experiments. We also appreciate your thoughtful suggestion regarding the naming and will deeply consider adopting "trajectory-based EPO" to better reflect our contribution.

---

### Meta-Review · Area_Chair_pTey · 2026-01-09

**Summary:**

The reviewers raised several significant concerns regarding the paper "Exploring the Entropy Mechanism in LLM Agents On-policy Optimization." Key points of contention included the adequacy of baselines, the clarity and accuracy of the method's naming, the theoretical rigor of the proposed framework (particularly concerning differentiability of the loss function, definition of theoretical constants, and assumptions about Lipschitz continuity), the novelty of certain components, and the empirical validation of the core "cascade failure" phenomenon and the proposed solution. Reviewer nuP9 initially identified a "fatal flaw" regarding the zero gradient of Equation (6), while Reviewer rz4m provided a detailed critique of the theoretical foundations. Reviewers 6TKC and U8i5 focused more on experimental comparisons and the conceptual understanding of the exploration-exploitation dilemma in the context of the proposed method.

**Reviewer Concerns:**

The authors provided comprehensive rebuttals that successfully addressed several critical concerns. Reviewer 6TKC's questions regarding baselines, hyperparameter effects, and the decaying entropy penalty were thoroughly addressed by pointing to existing experiments in the paper's ablation and model studies, leading to the reviewer's satisfaction. Reviewer U8i5's concerns about potential stifling of exploration and the measurement of accumulated uncertainty were also well-addressed by the authors, who clarified the use of soft bounds, dynamic anchoring, and adaptive weighting, and directed the reviewer to specific figures demonstrating the measurement of uncertainty propagation. Most notably, the authors provided a crucial clarification to Reviewer nuP9 regarding the "zero gradient" issue in Equation (6), explaining that their implementation uses a differentiable hinge loss (ReLU) formulation, thereby resolving what was deemed a "fundamental and fatal flaw" and updating the manuscript accordingly. However, several of Reviewer nuP9's other significant concerns remain largely outstanding, including the perceived lack of novelty, the heuristic nature of the βk design despite the authors' defense, the triviality of episodic entropy computation, and the need for more rigorous justification and specific empirical evidence for the link between entropy fluctuations and cascade failures beyond what was presented.

**Reviewer Scores:**

Based on the discussion and rebuttals, I anticipate a positive shift in scores for two of the reviewers, while the third might still remain critical. Reviewer 6TKC, who initially rated the paper a 4 (marginally below acceptance), explicitly stated they "would raise [their] score" after the clarifications. Given that all their major comments were addressed to their satisfaction, I would expect their score to increase to at least a 6 (weak accept). Reviewer U8i5, also starting at a 4, received detailed and direct responses to all their questions, often referencing existing parts of the paper. While they did not provide a follow-up comment, the thoroughness of the authors' rebuttal suggests that their concerns were likely alleviated, leading to an probable increase to a 6. Reviewer nuP9, who rated the paper a 2 (reject) due to multiple strong criticisms, including a "fatal flaw," received a crucial clarification on the mathematical implementation. This clarification resolves the most severe technical issue. However, given the reviewer's deep-seated skepticism regarding novelty, theoretical grounding, and empirical justification, it is unlikely that this single correction would lead to a full acceptance. Nevertheless, removing the "fatal flaw" should prevent an outright rejection, so I would predict their score to modestly increase to a 4 (marginally below acceptance), reflecting that while the paper is no longer fundamentally broken, significant weaknesses in contribution and justification remain in their view.

---

### Decision · Program_Chairs · 2026-01-26

Reject